# Ice cloud microphysical trends observed by the Atmospheric Infrared Sounder

Brian H. Kahn[1], Hanii Takahashi[1,2], Graeme L. Stephens[1], Qing Yue[1], Julien Delanoë[3], Gerald Manipon[1], Evan M. Manning[1], and Andrew J. Heymsfield[4]

[1]Jet Propulsion Laboratory, California Institute of Technology, Pasadena, CA, 91109, USA
[2]Joint Institute for Regional Earth System Science and Engineering, University of California – Los Angeles, Los Angeles, CA, 90095, USA
[3]LATMOS/IPSL, UVSQ-CNRS-UPMC, 11 Boulevard D'Alembert, 78280 Guyancourt, France
[4]National Center for Atmospheric Research, Boulder, CO, 80301, USA

*Correspondence to*: Brian H. Kahn (brian.h.kahn@jpl.nasa.gov)

**Abstract.** We use the Atmospheric Infrared Sounder (AIRS) version 6 ice cloud property and thermodynamic phase retrievals to quantify variability and 14-year trends in ice cloud frequency, ice cloud top temperature ($T_{ci}$), ice optical thickness ($\tau_i$) and ice effective radius ($r_{ei}$). The trends in ice cloud properties are shown to be independent of trends in information content and $\chi^2$. Statistically significant decreases in ice frequency, $\tau_i$, and ice water path (IWP) are found in the SH and NH extratropics, but trends are of much smaller magnitude and statistically insignificant in the tropics. However, statistically significant increases in $r_{ei}$ are found in all three latitude bands. Perturbation experiments consistent with estimates of AIRS radiometric stability fall significantly short of explaining the observed trends in ice properties, averaging kernels, and $\chi^2$ trends. Values of $r_{ei}$ are larger at the tops of opaque clouds and exhibit dependence on surface wind speed, column water vapour (CWV) and surface temperature ($T_{sfc}$) with changes up to 4–5 µm but are only 1.9% of all ice clouds. Non-opaque clouds exhibit a much smaller change in $r_{ei}$ with respect to CWV and $T_{sfc}$. Comparisons between DARDAR and AIRS suggest that $r_{ei}$ is smallest for single-layer cirrus, larger for cirrus above weak convection, and largest for cirrus above strong convection at the same cloud top temperature. This behaviour is consistent with enhanced particle growth from radiative cooling above convection or large particle lofting from strong convection.

## 1 Introduction

While our understanding of ice cloud microphysics has greatly advanced from targeted in situ campaigns during the past several decades (Baumgardner et al., 2017), global distributions obtained from satellite observing systems remain highly uncertain (e.g., Stubenrauch et al., 2013). Climate GCM simulations of the radiative response of high clouds have a large inter-model spread (e.g., Zelinka et al., 2013). Uncertainty in the ice hydrometeor fall speed is cited as a leading contributor despite dozens of modelling parameters contributing to this spread (e.g., Sanderson et al., 2008). The ice particle size

distribution (PSD) is closely correlated to the magnitude of fall speed (Heymsfield et al., 2013; Mitchell et al., 2008). The PSD is frequently summarized as an ice effective radius ($r_{ei}$) (McFarquhar and Heymsfield, 1998) in most publicly available satellite remote sensing data sets.

Global decadal-scale satellite-based estimates of $r_{ei}$ are available from two algorithm versions of the Moderate Resolution Imaging Spectroradiometer (MODIS) (Platnick et al., 2017; Minnis et al., 2011) and the Atmospheric Infrared Sounder (AIRS) (Kahn et al., 2014). Another estimate of $r_{ei}$ is derived from a combination of the Cloud-Aerosol Lidar with Orthogonal Polarization (CALIOP) and the Imaging Infrared Radiometer (IIR) instruments (Garnier et al., 2013) on the Cloud-Aerosol Lidar and Infrared Pathfinder Satellite Observation (CALIPSO) satellite (Winker et al., 2010). Additional active-passive algorithms are also available. Deng et al. (2013) describe comparisons of $r_{ei}$ between the CloudSat level-2C ice cloud property product (2C-ICE), radar-lidar profiles of $r_{ei}$ (DARDAR; Delanoë and Hogan, 2010), and the CloudSat level-2B radar-visible optical depth cloud water content product (2B-CWC-RVOD).

Satellite retrievals of $r_{ei}$ have known sources of uncertainty that are traced to the use of 1-D radiative transfer theory (e.g., Fauchez et al., 2015), errors that arise from a lack of precision, accuracy, or incomplete sampling in the atmospheric and surface state (e.g., Kahn et al., 2015), variability in mixtures of ice crystal habits and PSDs within a single satellite pixel (e.g., Kahn et al. 2008; Posselt et al., 2008), systematic uncertainties and approximations taken in the forward model (e.g., Wang et al., 2016; Irion et al., 2018), and instrument calibration drift (e.g., Pagano et al., 2012; Yue et al., 2017a; Manning and Aumann, 2017). The vertical heterogeneity of ice water content (IWC) and $r_{ei}$ are known to cause differences in $r_{ei}$ derived from shortwave/near-infrared (SWIR) and mid-infrared (MIR) bands for identical clouds and observing geometry (Zhang et al., 2010). Kahn et al. (2015) compared pixel-scale retrievals between MODIS Collection 6 (C6; Platnick et al., 2017) and AIRS Version 6 (V6; Kahn et al., 2014) and confirmed, for a small subset of homogeneous clouds, that $r_{ei}$ is typically 5–10 μm larger when derived from MODIS SWIR bands compared to AIRS MIR bands.

Secular trends in water path, cloud amount, and cloud height (e.g., Wylie et al., 2005; Dim et al., 2011; Bender et al., 2012; Marvel et al., 2015; Norris et al., 2016; Manaster et al., 2017) suggest that climate change signals may be observable within the satellite era. There is a notable absence of published studies regarding trends in ice microphysics. Sherwood (2002) inferred regional decreases of $r_{ei}$ by ~0.5 μm per decade using offline calculations of Advanced Very High Resolution Radiometer (AVHRR) radiances and argued for the importance of aerosols on cloud microphysics. Chen et al. (2016) describe experiments with the Nonhydrostatic ICosahedral Atmospheric Model (NICAM) that suggest an increase of $r_{ei}$ in a warming climate that may be related to a weaker tropical circulation, and therefore implies changes in the dominant pathways of ice nucleation, growth, and precipitation processes. The rapid intensification of Earth's hydrological cycle is consistent with increased convective aggregation (Mauritsen and Stevens, 2015), the narrowing and intensification of the ITCZ (Su et al., 2017), and a lower end estimate of climate sensitivity; the high cloud response is key to reconcile a strong hydrological response and lower end climate sensitivity (e.g., Lindzen et al., 2001; Mauritsen and Stevens, 2015). The areal extent of cloud anvils are correlated to upper tropospheric lapse rate (Bony et al., 2016) and additionally ice hydrometeor fall speed, with larger (smaller) anvils for smaller (larger) $r_{ei}$ (Satoh and Matsuda, 2009). The radiometrically stable AIRS

instrument (Pagano et al., 2012) may provide constraints on ice cloud microphysical properties that are highly desired (Kärcher, 2017).

While not the focus of this investigation, aerosols modulate ice clouds through numerous indirect and direct pathways (Liu et al., 2007; Gettelman et al., 2010; Kärcher, 2017). Variable magnitudes of $r_{ei}$ at convective tops result from variations in cloud condensation nuclei (CCN) or ice nuclei (IN); these responses are highly nonlinear because of complex liquid and ice phase microphysical processes as a function of updraft velocity, concentration and composition of CCN and IN (Phillips et al., 2007; Van Weverberg et al., 2013). Morrison and Grabowski (2011) used a 2-D cloud resolving model (CRM) to show that convection may subtly weaken yet cloud top height and anvil IWC may increase in polluted compared to clean conditions, resulting in a reduction of $r_{ei}$ by up to a factor of two. Simultaneous observations of $r_{ei}$ from MODIS and IWC from the Microwave Limb Sounder (MLS) suggest that more intense convection (defined by positive anomalies of IWC) increases $r_{ei}$ while polluted air (defined by positive anomalies of aerosol $\tau$) decreases $r_{ei}$ (Jiang et al., 2011). Whether convection is dynamically invigorated by increased aerosol concentrations, however, is highly debatable (Rosenfeld et al., 2008; Fan et al., 2013; Grabowski, 2015). Improving our understanding of satellite-based ice cloud microphysics will lay the groundwork for future observational constraints of aerosol-ice microphysics interactions.

Stanford et al. (2017) show that the covariability of cloud temperature, vertical velocity, and $r_{ei}$ observations within tropical mesoscale convective systems (MCSs) reveal very complex behaviour taken during the High Altitude Ice Crystals-High Ice Water Content (HAIC-HIWC) field campaign near Australia. HAIC-HIWC observations and simulations with the Weather Research and Forecasting (WRF) model using multiple microphysical parameterizations are compared. The observations show that larger (smaller) $r_{ei}$ are found in weaker (stronger) updrafts of MCSs, while the reverse is generally true for IWC; however the opposite behaviour was found for observations taken within a tropical cyclone (Leroy et al., 2017). The covariability of $r_{ei}$ with cloud temperature and vertical velocity among the WRF experiments strongly disagree in sign and magnitude compared to HAIC-HIWC observations.

The aforementioned observational, theoretical, and numerical modelling studies motivate the development of additional constraints on $r_{ei}$ and its covariability with other ice cloud, thermodynamic, and dynamic fields. This investigation is a first attempt to quantify secular changes in $r_{ei}$ from AIRS over its decade and a half observational record with well-characterized radiometric stability (Pagano et al., 2012). We attempt to identify potential algorithmic, information content, calibration, and sampling characteristics that induce nonphysical trends. Collocated AIRS and Advanced Microwave Sounding Unit (AMSR) data are used to glean insight among the connections between $r_{ei}$ and tropical convection/precipitation processes. Lastly, DARDAR data is used to investigate increases in $r_{ei}$ that occur near the tops of deep convection. The collocation of pixel-scale data among AIRS, AMSR, and DARDAR is a first step towards illuminating the potential processes that may be responsible for secular changes in ice cloud properties.

## 2 Data

### 2.1 Atmospheric Infrared Sounder (AIRS)

The AIRS V6 cloud properties are used from 01 September 2002 until 31 August 2016 (Kahn et al., 2014; K14 hereafter). As this investigation addresses ice microphysics, we focus on the 26.5% of AIRS pixels containing ice thermodynamic
phase, and their ice cloud top temperature ($T_{ci}$), ice optical thickness ($\tau_i$) and $r_{ei}$ retrieval parameters (K14). The identification of cloud thermodynamic phase is described in K14 and is validated against CALIOP phase estimates in Jin and Nasiri (2014). A set of four brightness temperature ($T_b$) thresholds and $T_b$ differences ($\Delta T_b$) in the 8–12 μm atmospheric window region identify ice phase by leveraging the spectral dependence of the refractive index of ice; the ice phase is identified correctly in excess of 90% of the time using CALIOP as truth. Ice clouds of convective origin typically have larger $\tau_i$
(Krämer et al., 2016) and trigger more of the ice phase tests, while tenuous ice clouds have smaller $\tau_i$ and trigger fewer phase tests.

For each AIRS footprint identified as containing ice, the three-parameter optimal estimation (OE) ice cloud property retrieval is performed (K14) and minimizes the following cost function:

$$C = \|\boldsymbol{y} - \boldsymbol{F}(\boldsymbol{x}, \boldsymbol{b})\|^2_{S_\varepsilon^{-1}} + \|\boldsymbol{x} - \boldsymbol{x_a}\|^2_{S_a^{-1}} \,, \tag{1}$$

where $\boldsymbol{F(x)}$ is the radiance vector that is forward modelled, $\boldsymbol{x}$ is the state vector of retrieved parameters, $\boldsymbol{b}$ is the vector containing fixed state parameters, $\boldsymbol{y}$ is the vector of AIRS radiances (K14), $\boldsymbol{x_a}$ is the prior guess of $\boldsymbol{x}$, $\boldsymbol{S_a^{-1}}$ is the inverse of the a priori covariance, and $\boldsymbol{S_\varepsilon^{-1}}$ is the inverse of the noise covariance of AIRS radiances. The retrieval state vector $\boldsymbol{x}$ is restricted
to $\tau_i$ (ice_cld_opt_dpth in L2 Support file; defined at 0.55 μm), $r_{ei}$ (ice_cld_eff_diam in L2 Support file; $D_e$ is converted to $r_{ei}$ henceforth), and $T_{ci}$ (ice_cld_temp_eff in L2 Support file). The bulk ice scattering models, and the definition of ice $D_e$, follows from Baum et al. (2007, cf. Eqn. 4). The surface properties (temperature and emissivity) and atmospheric profiles (temperature and specific humidity) in $\boldsymbol{b}$, and $\boldsymbol{x_a}$ (e.g., upper level cloud top temperature $T_{cld}$ used as a prior guess for $T_{ci}$) are taken from the AIRS cloud-clearing product (L2 Standard file AIRX2RET for IR/MW and AIRS2RET for IR only). The
averaging kernel matrix $\boldsymbol{A}$ quantifies the sensitivity of the retrieval with respect to changes in the true state:

$$A = (K^T S_\varepsilon^{-1} K + S_a^{-1})^{-1} K^T S_\varepsilon^{-1} K \tag{2}$$

Scalar averaging kernels (AKs; ice_cld_opt_dpth_ave_kern, ice_cld_eff_diam_ave_kern, and ice_cld_temp_eff_ave_kern in
L2 Support file) are reported for each of the three state vector retrieval parameters as off-diagonal terms of $\boldsymbol{A}$ in Eqn. (2) are not considered in K14. The scalar AKs quantify the information content of the retrieval with respect to $\boldsymbol{x}$. The normalized $\chi 2$ (ice_cld_fit_reduced_chisq in L2 Support file) is defined as:

$$\chi^2 = \frac{1}{N}\sum_{i=1}^{N}\left(\frac{y_i - [F(x)]_i}{\varepsilon_i}\right)^2 \quad , \tag{3}$$

where $\varepsilon_i$ is the radiance error in channel $i$ and N=59. The $\chi 2$ from Eqn. (3) is calculated for 59 observed and simulated 8-14

μm channels (K14) and is used to determine the robustness of the radiance fits.

We have also tested the fidelity of $r_{ei}$ within 6 K of the cold point tropopause. As AIRS observations are derived from IR thermal emission spectra, biases may arise in proximity to the tropopause. Any uncertainty in $T_{ci}$, the height and magnitude of the cold point tropopause, or temperature lapse rate may lead to increased uncertaintiy in $r_{ei}$ as small spectral changes translate to large geophysical retrieval changes. Therefore, "filtered" sets of AIRS retrievals that remove ice clouds within 6

K of the cold point tropopause are shown throughout the paper, then are compared with "unfiltered" data when contrasted to DARDAR retrievals (see Section 6).

The average 14-year climatology of $r_{ei}$, $\tau_i$, $T_{ci}$, and ice cloud frequency based on the IR only retrieval (AIRS2RET) is shown in Fig. 1. Retrievals with quality control (QC) flags QC=0 and QC=1 (K14; ice_cld_opt_dpth_QC, ice_cld_eff_diam_QC, and ice_cld_temp_eff_QC in L2 Support file) are included and the $\chi 2$ is filtered using the QC flag specific to $r_{ei}$. The

climatology is restricted to ice free ocean between 54°S–54°N as influences of surface heterogeneity including mountainous terrain, low emissivity such as bare mineral soils, and nocturnal and high latitude wintertime inversions on the ice parameter retrieval are not fully understood and warrant further investigation. A strict ad hoc QC may filter questionable data over land but the resulting data sample may skew the geophysical signals toward opaque clouds at the expense of non-opaque clouds that are more likely to be filtered out. The AIRS sensitivity is maximized for optically thinner cirrus with $\tau \leq 5$ (e.g., Huang

et al., 2004), while MODIS sensitivity is maximized for optically thicker cirrus (e.g., Kahn et al., 2015; Chang et al., 2017). The AIRS sampling includes nearly all ice clouds with $\tau_i > 0.1$, while the maximum values of $\tau_i$ asymptote to values near 6-8 (e.g., Kahn et al., 2015). The $r_{ei}$ is retrieved for the same sample although retrievals with QC=2 are not included. Kahn et al. (2015) describe pixel-level comparisons between AIRS and MODIS ice cloud properties and show that overlapping sensitivity for both $\tau_i$ and $r_{ei}$ is observed for optically thicker pixels containing four positive ice phase tests with spatial maps

resembling those described in King et al. (2013).

The well-documented (Wylie et al., 2005; King et al., 2013; Stubenrauch et al., 2013) spatial distributions of ice clouds with maxima in the tropics and extratropical storm tracks and minima in the subtropical gyres are shown in Fig. 1. Higher magnitudes of $r_{ei}$ are observed along the ITCZ where deep convection is more dominant, while lower magnitudes are observed in the western Pacific Warm Pool region where transparent cirrus is more dominant. These patterns are consistent

with previous observational and modelling studies. Yuan and Li (2010) found that MODIS $r_{ei}$ is a few μm larger at the tops of tropical deep convection when compared to the extratropics for the same cloud top temperatures and $T_b$s. Barahona et al. (2014) show an increase in $r_{ei}$ of a few μm in the Goddard Earth Observing System Model (GEOS-5) with improved comparisons against MODIS. Eidhammer et al. (2017) describe simulations using the Community Atmosphere Model

(CAM5) for a consistent single ice species across cloud and precipitating ice hydrometeors without thresholds for autoconversion. A subtle but distinct double ITCZ is observed in the zonal averages, not unlike AIRS $r_{ei}$ and $\tau_i$ in Fig. 1.

The loss of AMSU-A2 on 24 September 2016 led to the termination of the operational AIRS/AMSU combined infrared and microwave (AIRX2RET/AIRX2SUP) cloud clearing retrieval (Yue et al., 2017b). Since that time, the AIRS infrared only (AIRS2RET/AIRS2SUP) cloud clearing retrieval is the current operational version. The impacts of the loss of AMSU on retrieval parameters are assessed in great detail in Yue et al. (2017b). While many subtle changes were documented for vertical profiles of temperature and specific humidity, differences were more substantive for particular cloud properties. Further discussion regarding differences in Fig. 1 for AIRS2RET and AIRX2RET is found in the Appendix and shown in Fig. A1.

Figure 2 details AKs and $\chi^2$ using identical QC as Fig. 1. Typically the AKs are much lower and the $\chi^2$ fits much larger for QC=2 (not shown) and indicate lower information content and much poorer radiance fits. The AK and $\chi^2$ patterns are spatially coherent and exhibit small variations between tropical convection, thin cirrus, and both extratropical storm tracks. The AKs have maxima on the equatorial side of the storm tracks with a poleward minimum and additional minima within the tropics. Multi-layer clouds are more frequent in the tropics and extratropical storm tracks (Chang and Li, 2005; Mace et al., 2009) and lower values of $T_{ci}$ AKs in these areas is consistent with the single cloud layer assumption in the forward model (Kahn et al., 2015). However, the $r_{ei}$ AK distributions strongly suggest that the single layer assumption is providing high information content of ice cloud properties. Interestingly, the spatial patterns of $r_{ei}$ AKs are correlated to $\tau_i$ (compare Figs. 1 and 2) in the low latitudes, where large values of $\tau_i$ align with slightly reduced $r_{ei}$ AK. An area of reduced $r_{ei}$ AK in the subtropical north Atlantic corresponds to the Saharan air layer. Generally $\chi^2$ is about 3-4 between ±30° latitude with somewhat lower values (better fits) in the ITCZ and extratropics. The spatial pattern of $\chi^2$ does not closely track any of the AK patterns; thus for QC=[0,1] retrievals, spatial variations in information content do not correlate to the quality of the spectral fit. Differences between the IR/MW and IR only retrievals are generally minor but not negligible and are consistent with the loss of MW information. Further discussion regarding differences in Fig. 2 for AIRS2RET and AIRX2RET is found in the Appendix and depicted in Fig. A2.

**2.2 AMSR-E/AMSR-2**

Coincident satellite observations in the A-train are valuable for gaining physical insight about the AIRS ice cloud properties. The Advanced Microwave Sounding Unit–Earth Observing System (AMSR-E) on Aqua and AMSR-2 instrument on GCOM-W1 provide several atmospheric and surface properties that coincide with AIRS. The AMSR-E instrument was operational from 4 May 2002 until 4 October 2011, while AMSR-2 is currently operational since 18 May 2012. We focus on the summer months July and August from 2003–2016 in this investigation. The total column water vapour (CWV) cloud liquid water path (LWP), rain rate (RR) (Hilburn and Wentz, 2008), sea surface temperature ($T_{sfc}$) (Gentemann et al., 2010),

and direction-independent near surface wind speed (u) are obtained from AMSR-E (Wentz et al., 2014a) and AMSR-2 (Wentz et al., 2014b) using Version 7 data.

## 2.3 DARDAR

We use the CloudSat, CALIPSO, and MODIS radar/liDAR (DARDAR) combined ice cloud property retrieval developed by Delanoë and Hogan (2008; 2010) to investigate $r_{ei}$ in a variety of ice clouds. This algorithm is based on an OE algorithm that includes CALIOP's lidar backscatter and CloudSat's radar reflectivity to retrieve vertical profiles of IWC, $r_{ei}$, and other cloud variables. The ice cloud property retrievals are available at CloudSat's 1.4 km horizontal resolution and CALIPSO's 60 m vertical resolution and are matched to the nearest AIRS pixel. As the radar and lidar have different sensitivities to a range of cloud characteristics, the retrieval is functional when one of the instruments is unable to detect clouds either because of small ice particles or strong attenuation. Deng et al. (2013) showed that there is very good agreement in $r_{ei}$ between DARDAR, 2C-ICE, and in situ obtained observations from the Small Particles in Ice (SPARTICUS) field campaign; therefore only DARDAR data is used in this study.

DARDAR contains two products: DARDAR-MASK (Delanoë and Hogan, 2010; Ceccaldi et al., 2013) and DARDAR-CLOUD (Delanoë and Hogan, 2008, 2010), which are both available through the ICARE Thematic Center (http://www.icare. univ-lille1.fr/drupal/archive/). DARDAR-MASK provides the vertical cloud classification and a range of additional categorizations (e.g., clear, aerosols, rain, supercooled and warm liquid, mixed phase, ice). DARDAR-CLOUD provides ice cloud properties such as extinction, $r_{ei}$, and IWC by a variational radar-lidar ice-cloud retrieval algorithm called VarCloud (Delanoë and Hogan, 2008). In this product, normalized ice particle size distributions are used and non-spherical particles leverage in situ measurements (Delanoë et al., 2014).

## 3 Methodology

### 3.1 Pixel matching

A pixel scale nearest neighbour matching approach (Fetzer et al., 2013) is applied to AIRS/AMSU, AMSR-E and AMSR-2 from 01 September 2002 until 31 August 2016. By retaining the native spatial and temporal covariances in the matched data, smaller scale and faster temporal processes are captured that are otherwise lost to spatial gridding and temporal averaging (e.g., Kahn et al., 2017). The same methodology is applied between AIRS/AMSU and DARDAR from 01 July 2006 until 31 December 2008 within the subset of AIRS/AMSU pixels along the CloudSat-CALIPSO ground track.

### 3.2 Secular trends

Trends and their statistical significance are calculated following Santer et al. (2000) using a two-sided t-test and confidence intervals at the 95% significance level. The 95% confidence intervals are also calculated for a lag-1 autocorrelation in order to assess the sensitivity to highly correlated time series (Cressie, 1980; Santer et al., 2000). Trends are calculated for two

different spatial averages. First, global monthly anomalies at 1°×1° resolution over the 14-year observing period are calculated then trends are reported at the same spatial resolution. Second, monthly anomalies averaged between 54°S–18°S, 18°S–18°N, and 18°S–54°N are calculated then trends with 95% confidence intervals with and without lag-1 autocorrelation are determined and displayed as box and whisker diagrams. Three sets of AIRS properties are described: combined IR/MW, 5   IR only, and IR only for the 1/3 of pixels in the swath nearest to nadir view. The purpose of contrasting results between IR only and combined IR/MW is to highlight differences between the two algorithms with further detail found in the Appendix. The purpose of showing near nadir IR only is to demonstrate the overall lack of sensitivity in the results to scan angle.

### 3.3 Joint histograms

Instantaneous pixel matches of AMSR-E and AMSR-2 variables versus AIRS ice cloud properties are used to construct joint 10   histograms following the approach described in Kahn et al. (2017). The joint histograms contain the natural log of counts with $r_{ei}$ and $\tau_i$ contours superimposed. The histograms each contain one of several AMSR variables on the x-axis and the AIRS upper layer $T_{cld}$ on the y-axis. The intent of these diagrams is to reveal the physical response of cloud top $r_{ei}$ and $\tau_i$ (if any) to precipitating and non-precipitating cloud types and meteorological variability inferred from AMSR. The emphasis is on $r_{ei}$ versus $T_{cld}$ to facilitate comparisons with previous works that describe temperature dependence of $r_{ei}$. Convective and 15   non-convective cloud types are shown separately in order to highlight the much larger responses of $r_{ei}$ to thermodynamic and dynamical variability in tropical convection.

As discussed in Kahn et al. (2014), the $T_{ci}$ variable is included in the retrieval state vector to improve the $\chi^2$ radiance fits and the success rate of retrieval convergence. While there are strong similarities between $T_{ci}$ and the upper level $T_{cld}$, some differences arise within multi-layer clouds as expected since $T_{ci}$ is based on the assumption of a single-layer cloud (Kahn et 20   al., 2014). Further discussion on the reconciliation of the two cloud top temperatures is in progress and will be presented in a separate manuscript.

### 3.4 Comparing DARDAR and AIRS $r_{ei}$

The retrievals of $r_{ei}$ from AIRS and DARDAR are separately compared for single-layer cirrus and convective clouds over ocean. We use the CloudSat reflectivity, DARDAR-MASK, and DARDAR-CLOUD products to define cirrus only (Ci), 25   cirrus above weak convection (Weak Conv), and cirrus above strong convection (Strong Conv). We begin with identifying the number of cloud layers in each profile and the cloud top height (CTH) and cloud bottom height (CBH) of each layer. Profiles with more than three layers of clouds are not included in the statistics. Cirrus is defined as CBH>12km (e.g., Keckhut et al., 2006), and deep convection is defined as CTH>10km and CBH<2km (Takahashi and Luo, 2014). Among deep convective clouds, weak and strong convection are defined by the echo top height (ETH) of the 10 dBZ contour: the 30   ETH of 10dBZ>10km (Luo et al, 2008; Takahashi and Luo, 2012, 2017) for strong deep convection, while the ETH at 10dBZ<5km for weak deep convection. For the statistics described later, the upper layer cirrus is chosen to have 1)

CTH<tropopause height, 2) cloud base temperature<200K, and cirrus geometrical thickness <1km. As mentioned in Section 3.3, the coarse classification of convective intensity is useful for quantifying differences in cloud top $r_{ei}$ for non-convective, weakly, and strongly convective scenes.

## 4 Results

### 4.1 Global trends

The 14-year temporal trends in the cloud properties depicted in Fig. 1 are shown in Fig. 3. The $T_{ci}$ decreases about 1–2 K in most areas, but a few regions are found to increase. The trend in $r_{ei}$ is increasing most everywhere, from a few tenths to 1–2 µm, with the largest values in proximity to tropical convection. While the upward trend in $r_{ei}$ is somewhat noisy at 1°×1°, the increase is notably consistent across the global oceans. In contrast, $\tau_i$ is decreasing in most regions except for convectively active regions in the ITCZ, parts of the tropical western Pacific Warm Pool, and southern Indian Ocean. The trend in ice frequency is similar to $\tau_i$ but is spatially smoother and the magnitude is much larger in the tropics. We reiterate that the AIRS sensitivity to ice clouds is limited between $0.1 < \tau_i < {\sim}6\text{-}8$, thus the $\tau_i$ trends do not include contributions from clouds outside of this sensitivity range.

Trends in the AKs and $\chi^2$ are shown in Fig. 4. Generally speaking, the trends in the retrieval parameters do not resemble the spatial patterns of trends in their respective AKs. The trends for $r_{ei}$ AKs in Fig. 4 resemble $\tau_i$ trends in Fig. 3. This shows that changes in the information content of $r_{ei}$ are related to trends in $\tau_i$. The trends for $r_{ei}$ however do not resemble trends in either $\tau_i$ or $\tau_i$ AKs. We conclude that the upward trend in $r_{ei}$ is independent of the gain or loss of information in $r_{ei}$. The trends in $\chi^2$ are quite subtle and are smallest when the ice frequency is largest. There is an increased zonal symmetry in the $T_{ci}$ AK trend than the $T_{ci}$ trend in Fig. 3 and no obvious correspondence is apparent between the sign of the $T_{ci}$ trend in Fig. 3 and the $T_{ci}$ AK trend in Fig. 4. The $\tau_i$ AK trend is generally upwards except in a few tropical locations including the Saharan air layer. The spatial patterns in $\tau_i$ AK trends do not appear to be related to other fields. In conclusion, the observed trends in the ice cloud properties in Fig. 3 do not appear to be caused by the gain or loss of information content, or the fidelity of the radiance fitting in the retrieval.

### 4.2 Latitude bands

The anomaly time series of $r_{ei}$ for three latitude bands and the IR/MW, IR only, and IR nadir retrievals are shown in Fig. 5. While the anomaly time series for most of the 1°×1° grid boxes are not statistically significant (not shown), a statistically significant signal is obtained for the broad oceanic latitude bands. The strong similarity in the anomaly time series for IR/MW, IR only, and IR nadir is apparent for $r_{ei}$ and is also true for the other variables (not shown). Despite some of the regional spatial differences in IR/MW and IR only retrievals described in the Appendix, the algorithm differences are miniscule with regard to the latitude band anomaly time series.

Figure 6 summarizes the results for three latitudinal bands and IR/MW, IR only, and IR nadir including 95% confidence intervals with and without lag-1 autocorrelation. The results in Fig. 6 have clouds within 6 K of the tropopause filtered out; very similar results are obtained with no filtering, and no material changes in sign, magnitude, and statistical significance are found. The extratropical SH and NH upward trends in $r_{ei}$ (Fig. 6b) are steadier with time than the tropics, which appears initially flat but then jumps upwards since 2013 (Fig. 5); however all three regions and retrievals exhibit statistically significant trends (Fig. 6). The $T_{ci}$ trends (Fig. 6a) show about 0.5–1.0 K of cooling for the three latitude bands and three data sets, with IR nadir exhibiting 95% confidence except for the lag-1 autocorrelation. The $\tau_i$ trends (Fig. 6c) are downward by 0.025–0.075 (significant at 95% in the SH and NH, but not in the tropics). The AKs are slightly downwards for $T_{ci}$ (Fig. 6d) and $r_{ei}$ (Fig. 6e) and upwards for $\tau_i$ (Fig. 6f); there is a mix of statistical significance that depends on the retrieval algorithm and latitude band. The $\chi^2$ trends (Fig. 6h) are essentially not statistically significant except for IR nadir in the tropics, although the trend is slightly downwards. This is consistent with an increasing thermal contrast with time as SSTs increase and ice clouds slightly cool. Ice frequency (Fig. 6g) decreases by about 1% relative to the sum of all AIRS pixels in the SH and NH extratropics (significant at 95%) and much less so in the tropics (not significant at 95%). Lastly, the ice water path (IWP) (Fig. 6i), which is calculated from $(2/3)\rho_i r_{ei}\tau_i$, where $\rho_i$ is the density of ice (assumed to be 0.92 g cm$^{-3}$), decreased about 0.4–0.8 g m$^{-2}$ in the SH (marginal significance at 95%) and NH (significant at 95%), and increased about 0.25 g m$^{-2}$ in the tropics (not significant at 95%). This latitudinal pattern is similar to the CMIP5 zonal average ensemble mean (Ceppi et al., 2016; cf. Fig. 1c) although the decrease of AIRS IWP appears larger in magnitude than CMIP5 IWP at first glance. The statistical significance of IWP is reduced, however, when compared to $\tau_i$ because of the compensating increases in $r_{ei}$ at all latitudes.

## 4.3 Potential impacts from radiometric drift

While the radiometric stability of AIRS is established to be approximately 4 ±1 mK yr$^{-1}$ (Pagano et al., 2012), the impacts of radiometric drift on secular cloud property trends has not been assessed to date. Manning and Aumann (2017) raise the possibility of channel-dependent drift that could cause a systematic change in slope in the 8–14 μm region especially for cold scenes with $T_b$ < 200 K. However these effects are subtle and are undetermined (but likely much smaller) for warmer $T_b$s that dominate the vast majority of AIRS pixels.

Four different tests have been designed to estimate the potential effects of calibration drift on the cloud products: (1) add +50 mK to all 59 cloud retrieval channels in K14 after cloud clearing but before the ice cloud property retrieval; (2) add +50 mK to cloud clearing channels and K14 channels but before cloud clearing is performed; (3) increase the slope across the 59 channels by reducing the longest wavelength channel by -50 mK, increasing the shortest wavelength channel by +50 mK, and apply a linearly interpolated correction to channels in between; (4) the same as (3) except decrease the slope by reversing the radiometric perturbations; (5) difference ice cloud properties between v6 and a new version (v7j) of AIRS radiances that is updated with new calibration estimates (S. Broberg and T. Pagano, personal communication); and (6) the

same as (5) except restricted to the new polarization corrections only. Experiment (1) isolates radiometric drift on ice cloud properties only, while (2) accounts for impacts from the full geophysical retrieval. Experiments (3) and (4) are an approximate way to estimate channel dependent drift but ignores the effects of individual detector modules. Experiments (1)-(4) are performed for the focus day 06 September 2002. Experiment (5) assesses the differences in ice cloud properties solely

due to the updated radiance calibration (v7j) and experiment (6) isolates the contributions from updates in the polarization corrections only. Experiments (5) and (6) are performed for the focus days 06 September 2002, 03 March 2007, 06 June 2007, 09 December 2007, 01 September 2012, and 12 September 2017. Further investigation into radiometric drift scenarios on the AIRS Level 2 retrieval system that take into account individual characteristics of each relevant detector module is warranted.

The results of experiments (1) and (2) are summarized in Table 1 for one day of retrievals on 6 September 2002 for ±54° latitude over the oceans and for QC=[0,1]. Both experiments indicate that differences with respect to the IR/MW retrieval are substantially smaller than the trends reported in Fig. 6. For the experiments before and after cloud clearing, $\Delta r_{ei}$ is -0.13 and -0.1 μm, $\Delta \tau_i$ is +0.01 and +0.005, and $\Delta T_{ci}$ is +0.24 and +0.26 K, respectively. The AK trends for cloud variables for the perturbation experiments are much less than depicted in Fig. 6. Furthermore, the sign changes of the perturbations are not

consistent with Fig. 6. (A similar test with −50 mK that is not shown was performed and is nearly symmetric but with opposite sign.) The results of the slope perturbation experiments (3) and (4) are summarized in Table 2. The $\Delta r_{ei}$ and $\Delta \tau_i$ trends are somewhat larger than for experiments (1) and (2) including a few of the AK trends, but still fall short of the magnitudes reported in Fig. 6 (also listed in Table 2 for IR only). As with experiments (1) and (2), the signs of the trends are not in agreement between the observed trends and the slope perturbation experiments, suggesting that the slope adjustment

does not explain observed trends. While the four highly simplified radiance perturbation experiments fall short of explaining the sign and magnitude of the observed ice property trends, some fractional contribution to observed trends cannot be ruled out.

Experiment (5) shows that the differences between the standard v6 retrieval and that using v7j radiances are somewhat larger than the differences obtained from experiments (1)-(4) and depicted in Tables 1 and 2, and are closer to magnitudes obtained

in the 14-year trends depicted in Fig. 6 (not shown). However, the differences are nearly identical for all focus days listed above that are dispersed throughout the length of the AIRS mission. Therefore, the new v7j radiance calibration estimates indicate that the update will not cause meaningful changes in the sign and magnitude of ice cloud property trends. Rather, some small shift in the magnitude of the mean properties may occur that are similar in magnitude to the absolute value of the trends themselves. This behaviour holds for the left, center, and right 1/3 of the AIRS swath, although a very slight scan

dependence on the mean difference between v6 and v7j is observed. Lastly, experiment (6) shows that the polarization effects are about an order of magnitude smaller than experiment (5) (not shown). In summary, an eventual implementation of v7j (or similar) radiances should have no material impact on the secular trends derived in Fig. 6. However, this does not eliminate the possibility of a partial contribution from radiometric drift itself as described in experiments (1)-(4).

# 5 Insight into convective processes with AMSR

Using ground-based retrievals at Darwin, Protat et al. (2011) found that $r_{ei}$ is ~1–3 μm larger during active deep convection compared to suppressed conditions. Using a simultaneous retrieval of ice cloud properties from the MODIS and POLDER instruments, van Diedenhoven et al. (2014) found that $r_{ei}$ is larger in stronger convective events compared to others at a given cloud top pressure. van Diedenhoven et al. (2016) used airborne remote sensing observations to show that $r_{ei}$ decreases with height, reaches a minimum around 14 km, and may subsequently increase at higher altitudes. Barahona et al. (2014) obtained an increase of a few μm for clouds colder than 195 K with GEOS-5 and Hong and Liu (2015) show similar results with DARDAR data for the thickest convective clouds above 10 km. Convective overshoots into the lower stratosphere are 0.3–4.0 μm larger than non-overshooting deep convection observed with DARDAR data (Rysman et al., 2017). Lawson et al. (2010) use in situ aircraft probe data of ice particles in aged cirrus in proximity to convection to show that particle size is more weakly dependent on temperature below 215 K. Collectively, the aforementioned investigations suggest that $r_{ei}$ varies significantly at the tops of convective ice clouds and motivates the synergistic use of AMSR and AIRS at the pixel scale to capture convective-scale processes.

## 5.1 Differences in $r_{ei}$ between opaque, non-opaque, and multi-layer clouds

We show separate vertical profiles of AIRS $r_{ei}$ with ice $T_{cld}$ for opaque, non-opaque, and multi-layer clouds in Fig. 7 with definitions listed in Table 3. The two-layer effective cloud fraction (ECF) product is used to categorize the three cloud types and the threshold of 0.98 used by Protopapadaki et al. (2017) is applied to the upper layer to categorize opaque clouds. The ECF is a cloud product that represents the convolution of cloud fraction and cloud emissivity. Nasiri et al. (2011) showed that ECF from AIRS and effective emissivity from MODIS is in excellent agreement for both single and multi-layered cloud configurations. Only 1.9% of cloud tops are identified as opaque, and 98.1% are classified as non-opaque or multi-layer. The three categories of clouds are further subdivided into three additional categories using AMSR-E/AMSR-2 LWP and RR (Fig. 7): no liquid cloud or rain (LWP=0 and RR=0), liquid cloud but no rain (LWP>0 and RR=0), and liquid cloud with rain (LWP>0 and RR>0). Symbols show mean $r_{ei}$ over 10 K bins derived from IWC and extinction observations for three tropical in situ field campaigns: the Cirrus Regional Study of Tropical Anvils and Cirrus Layers–Florida Measurements for Area Cirrus Experiment (CRYSTAL-FACE, diamonds), the Tropical Composition, Cloud, and Climate Coupling Experiment (TC4, triangles), and the NASA African Monsoon Multidisciplinary Analysis (NAMMA, squares) campaigns (e.g., Heymsfield et al., 2014).

A maximum $r_{ei}$ occurs near 230 K and decreases at warmer and colder $T_{cld}$ for non-opaque and multi-layer clouds in the tropics (Fig. 7).  About 23.0% (17.3%) of cases are clear for non-opaque (multi-layer) according to AMSR but cloudy according to AIRS (Table 4). About 62.7% (53.3%) have LWP>0 and RR=0 for non-opaque (multi-layer) clouds. Both of these categories have nearly identical $T_{cld}$ dependence but multi-layer is several μm larger than non-opaque.  The remaining 14.3% (29.4%) have LWP>0 and RR>0, and $r_{ei}$ is 0.5-2.5 μm larger than no rain, with the largest differences for $T_{cld}$ <210K.

For opaque clouds in the tropics, $r_{ei}$ is substantially larger for all $T_{cld}$ than non-opaque clouds. There is three times the relative frequency of occurrence with raining opaque clouds compared to raining non-opaque clouds. An increase in $r_{ei}$ is found for increasing $T_{cld}$ unlike non-opaque and multi-layer clouds. A weaker vertical dependence in $r_{ei}$ for $T_{cld}$ <210K occurs and is similar to the results of van Diedenhoven et al. (2016). While this region is sensitive to the 6 K cut-off used to filter questionable retrievals near the tropopause, the profile in Fig. 7a is fairly robust for ~4 K and larger. While the in situ $r_{ei}$ estimates are somewhat larger for $T_{cld}$ > 220 K, this is expected as these observations may be obtained well below cloud top in which the infrared signal is not sensitive. Furthermore, the standard deviation of the in situ observations (not shown) has a magnitude that is approximately as large as the AIRS estimates of standard deviation, providing significant overlap between satellite and in situ observations.

## 5.2 Dependence of $r_{ei}$ on near surface wind speed

Correlations between rain rate and near surface wind speed in passive MW observations have suggested a tropical precipitation-convergence feedback (Back and Bretherton, 2005). The AMSR low frequency (LF) wind is used to quantify the response of $r_{ei}$ to variability in near surface wind speeds in the presence of convection. Figure 8 shows that opaque clouds exhibit dependence on wind speed; the weakest (strongest) winds are associated with the largest (smallest) $r_{ei}$. The change in $r_{ei}$ is ~2 μm for $T_{cld}$<210K but can be as large as 3–5 μm for $T_{cld}$>230 K. The dependence for non-opaque clouds is of opposite sign and lower magnitude when compared to opaque clouds. Figure 8 was also calculated with NWP model winds and is nearly identical with <1 μm difference (not shown). The consistency between NWP and AMSR winds provided confidence to partition the NWP winds into the individual u- and v-components, and the u-component is shown in the left column of Fig. 8. The largest values of $r_{ei}$ are found for light easterly winds around 5–10 m s$^{-1}$ for opaque, consistent with convectively active regimes generating larger $r_{ei}$ (e.g., Protat et al., 2011). Smaller values of a few μm are associated with westerly winds during suppressed convection. Changes in $r_{ei}$ due to wind direction changes are largest for $T_{cld}$>230K and are lowest for the coldest convective cloud tops. Very weak dependence is found for non-opaque clouds.

Multi-layer clouds exhibit the largest changes with wind speed (Fig. 8). However, the reduced values of $r_{ei}$ at higher wind speeds have low occurrence frequencies (i.e., noted by the gray scale shading). The contribution of retrieval biases that arise from an additional lower layer(s) not accounted for in the forward model (Kahn et al., 2014) has not been quantified. A firm conclusion on the realism of changes in multi-layer cloud top $r_{ei}$ to wind speed variability thus remains elusive and warrants further investigation.

The response of $\tau_i$ to near surface wind speed is shown in Fig. 9 for the same cloud categories in Fig. 8. There is little, if any, dependence of $\tau_i$ on wind speed except for slightly larger values during light easterly winds. The values of $\tau_i$ are lowest for non-opaque and confirm that these clouds are typically thin cirrus. The values of $\tau_i$ are a few factors larger for multi-layer and increase for stronger winds.

## 5.3 Dependence of $r_{ei}$ on CWV and $T_{sfc}$

The $T_{sfc}$ and CWV also exhibit correlations with $r_{ei}$ (Fig. 10). Opaque clouds exhibit a 2–4 μm jump as CWV increases from 45-65 mm, with the largest values for $T_{cld}>230K$; a similar pattern is observed for $T_{sfc}$ (Fig. 10). Little dependence of $r_{ei}$ is observed for non-opaque clouds. There is an increase in $r_{ei}$ with increasing CWV and $T_{sfc}$ for multi-layer clouds. Overall the $r_{ei}$ is lower in non-opaque clouds compared to opaque clouds for all combinations of $T_{ci}$, CWV, and $T_{sfc}$. There is a subtle dependence of $\tau_i$ on CWV for opaque clouds, although this dependence is absent in $T_{sfc}$ (Fig. 11). Both non-opaque and multi-layer clouds show increases in $\tau_i$ with CWV, but this only holds true for multi-layer clouds for increasing $T_{sfc}$.

## 6 Comparisons of AIRS and DARDAR $r_{ei}$

In the previous section, we showed that AIRS observes larger $r_{ei}$ at the tops of convection, and variations that depend on surface wind speed, $T_{sfc}$, CWV and precipitation rate. In this section, the $r_{ei}$ at the base of single-layer cirrus clouds, cirrus clouds above weak deep convection, and cirrus clouds above strong deep convection will be shown separately using the classification defined in Section 3.4. Stephens (1983) argued for a radiative mechanism that leads to $r_{ei}$ growth in cirrus overlying lower-layer clouds from enhanced radiative cooling; particle growth (decay) occurs in a radiatively cooled (heated) environment. DARDAR data is used to test whether the mechanism of Stephens (1983) is operating in observations, and also to determine the behaviour of AIRS in the same clouds. As opacity increases, or the higher (colder) the lower layer cloud occurs, the cooling from the cirrus layer is enhanced and larger $r_{ei}$ should therefore be observed. Systematic changes in the global circulation and changes in convective clustering and cloud overlap may lead to a higher frequency of overlapping cirrus on top of convection, and a reduced frequency of thin cirrus with climate change evidenced by trends in the Multi-angle Imaging SpectroRadiometer (MISR) derived cloud texture (Zhao et al., 2016). Thus, we will assess if this mechanism is a viable contributor to upward trends in $r_{ei}$.

Figure 12 shows the median (centre lines), mean (asterisks), and interquartile range (bottom and top edges of boxes) for DARDAR, IR/MW, and IR only AIRS retrievals, with filtered (non-filtered) versions of IR/MW and IR only that have retrievals within 6 K of the cold point tropopause removed (retained). For single-layer cirrus (Ci), AIRS is typically 2-3 μm larger than DARDAR with about 1 μm difference between IR/MW and IR only. For cirrus above weak convection (Weak Conv), AIRS remains about 2-3 μm larger than DARDAR as all AIRS retrievals increase about 2-3 μm in the mean compared to single-layer cirrus. For cirrus above strong convection (Strong Conv), the differences between DARDAR and AIRS are reduced, with IR only about 1-1.5 μm larger than IR/MW. The width of the DARDAR quantiles increases from single-layer cirrus to cirrus over strong convection indicating increased variability in $r_{ei}$; AIRS and DARDAR have nearly identical variability for cirrus above strong convection. AIRS and DARDAR exhibit an increase in $r_{ei}$ as convective cloud appears below cirrus layers suggesting that the mechanism described in Stephens (1983) may be operating in these differences. Partitioning contributions to the increase in $r_{ei}$ from the Stephens (1983) radiative cooling mechanism for

particle growth, and lofting of large ice particles from adjacent convection, however, cannot be easily differentiated; further investigation is warranted in the context of $r_{ei}$ trends shown in this work.

**7 Summary, Conclusions, and Outlook**

While high-level ice clouds are a key component of the changing climate system, there remains an absence of well-characterized observational constraints of ice cloud microphysics that are desired for climate model evaluation (Kärcher, 2017). The Atmospheric Infrared Sounder (AIRS) instrument, launched in May of 2002, is radiometrically stable within ±3-4 mK yr$^{-1}$ (Pagano et al., 2012). We use the AIRS version 6 ice cloud property and thermodynamic phase retrievals (Kahn et al., 2014) to quantify variability and trends in ice cloud frequency, ice cloud top temperature ($T_{ci}$), ice optical thickness ($\tau_i$) and ice effective radius ($r_{ei}$) from 01 September 2002 until 31 August 2016. We also investigate the scalar averaging kernels (AKs) associated with each retrieval quantity to determine changes in information content; and the $\chi^2$ fitting parameter, which quantifies the fidelity of the observed and simulated radiance fits across cloud types. Differences are described between in the ice cloud properties using the AIRS/AMSU combined cloud-clearing retrieval (IR/MW), the AIRS only cloud-clearing retrieval (IR only), and a subset of IR only for the 1/3 of pixels nearest to nadir view (IR nadir).

Spatial patterns in $r_{ei}$ reflect differences in proximity to deep convection, thin cirrus, and extratropical storm tracks. The averaging kernels and $\chi^2$ patterns are spatially coherent and exhibit variations between different cloud regimes with slight variations observed. The spatial patterns of $\chi^2$ and averaging kernels do not resemble each other. For the highest quality retrievals, we can conclude that the spatial variations in information content do not correlate to the spectral radiance fit in the retrievals of Kahn et al. (2014). The trends are nominally 1–2 K of cooling in $T_{ci}$, an increase of a few tenths to ~1 μm in $r_{ei}$, with the largest values in proximity to tropical convection. In contrast, $\tau_i$ is decreasing except for regions with high frequencies of convection (ITCZ, W. Pacific Warm Pool, and S. Indian Ocean). The trend in ice frequency is similar to $\tau_i$ but is spatially smoother and the magnitude is much larger.

The statistical significance of three wide latitude bands in the tropics, SH, and NH extratropics are calculated following the methodology of Santer et al. (2000). Ice frequency decreases by about 1% with respect to the total frequency of all AIRS pixels in the SH and NH extratropics (significant at 95%), and much smaller in the tropics (not significant at 95%). The ice water path (IWP) decreases by 0.4–0.8 g m$^{-2}$ in the SH (marginal significance at 95%) and NH (significant at 95%), and increases about 0.25 g m$^{-2}$ in the tropics (not significant at 95%). The statistical significance of IWP is lower than $\tau_i$ because of compensating increases in $r_{ei}$ at all latitudes (significant at 95%). Impacts of assumed radiometric drift on cloud property trends were determined for a few perturbation experiments. The perturbation experiments fall significantly short of explaining the magnitude of the observed ice property, AKs, and $\chi^2$ trends, although some fractional contribution to trends cannot be ruled out.

Surface based and aircraft in situ observations have demonstrated that active periods of deep convection are associated with larger $r_{ei}$ at cloud top (e.g., Protat et al.; 2011; van Diedenhoven et al., 2014). Values of AIRS $r_{ei}$ are plotted against $T_{cld}$ for opaque, non-opaque, and multi-layer clouds separately; only 1.9% of ice cloud tops are identified as opaque. Values of $r_{ei}$ are 3–9 µm larger (depending on cloud top temperature) for opaque clouds that are treated as a proxy for deep convection. A weaker dependence of $r_{ei}$ with $T_{cld}$<210K occurs and shows similarity to van Diedenhoven et al. (2016). Opaque clouds exhibit some dependence of $r_{ei}$ on wind speed. Opaque clouds also exhibit a 2–4 µm jump as CWV increases from 45-65 mm, and a similar pattern is observed for $T_{sfc}$. Non-opaque clouds do not exhibit much of a dependence on CWV and $T_{sfc}$.

The differences in $r_{ei}$ at the tops of opaque, non-opaque and multi-layer ice clouds due to thermodynamic and dynamic variability suggest distinct cloud regime behaviour. The low frequency of occurrence of opaque ice clouds (1.9%) implies that temporal changes in deep convective behaviour are likely overwhelmed by non-opaque and multi-layer behaviour. Secular changes in the joint histograms themselves, and furthermore changes in the frequency of occurrence within the joint histogram bins, must be quantified to deduce if trends or variability in CWV, rain occurrence, surface wind speed, and other relevant geophysical parameters such as low-level convergence (e.g., Stephens et al., 2018) could explain in part or whole secular trends in $r_{ei}$.

Comparisons between DARDAR and AIRS $r_{ei}$ are made for single-layer cirrus, cirrus above weak convection, and cirrus above strong convection. AIRS is typically 1-3 µm larger than DARDAR with about 1 µm difference between IR/MW and IR only. AIRS and DARDAR exhibit an increase in $r_{ei}$ as convective cloud appears below cirrus layers suggesting that the mechanism described in Stephens (1983) may be operating in these differences. However, secular trends in convective aggregation, convective mode, the probability distribution of vertical velocity, and ice nucleation and growth mechanisms that may change within the changing character of convection (e.g., Chen et al., 2016), are highly complex and uncertain in observations. The pixel-level collocations of AIRS, AMSR, and DARDAR are a first attempt at identifying atmospheric processes that could be responsible for secular trends in ice cloud properties. Further research is necessary to quantify the links between trends in ice cloud microphysics shown in this work with cloud responses in climate model simulations.

Current and future work will require the full AIRS and AMSR observational record to be collocated at the pixel scale to derive secular trends within the joint histograms. This will enable the assessment of whether there are preferred cloud, thermodynamic, or dynamical regimes that exhibit either strong or weak trends, or perhaps whether opposing trends among regimes exist. Further research is necessary to determine if retrieval biases may explain in part the variability exhibited in the multi-layer cloud category. The results presented herein were restricted to the July-August time period and ±18° latitude oceans. Cursory inspection of other latitudes and months (not shown) can exhibit different behaviour and may indicate fundamental differences in time changes of ice cloud properties between tropical and extratropical cloud processes. Lastly, this investigation does not include the analysis on cloud thermodynamic phase differences (liquid versus ice), cloud top temperature, and ECF, as those results will be reported elsewhere.

**Appendix**

Figure A1 shows that the AIRS algorithm differences in IR only (AIRS2RET) minus IR/MW (AIRX2RET) on the frequency of ice phase clouds over the global oceans are less than 1%. This implies that virtually the same sets of pixels are identified as ice in the two retrievals. The impacts on $T_{ci}$ are on the order of 1–2 K difference with IR only slightly cooler especially in the tropics. Some subtle and spatially coherent changes in $\tau_i$ of ~0.1 are observed with IR only thinner. The $r_{ei}$ is larger by 0.5–1.0 μm in IR only compared to IR/MW but a few of the extratropical storm track regions exhibit an opposite sign. To summarize, for the IR only retrieval, $\tau_i$ is lower by 0.1, $r_{ei}$ is larger by 0.5–1.0 μm in most areas, and $T_{ci}$ is lower by 1–2 K depending on the latitude.

Figure A2 shows the same differences as Fig. A1, except for the $T_{ci}$, $r_{ei}$, and $\tau_i$ AKs, and the $\chi^2$ fitting parameter. The differences in AKs and $\chi^2$ are generally small but some notable regional differences are observed. The $T_{ci}$ AK is mostly lower for the IR only compared to IR/MW and is consistent with added information provided by the microwave channels in the AMSU instrument. The $r_{ei}$ AK is larger for the IR only compared to IR/MW but the difference is an order of magnitude smaller than the $T_{ci}$ AK difference. The $\tau_i$ AK difference is small and does not exhibit coherent cloud-regime dependence except for the Saharan air layer. The difference in $\chi^2$ shows slightly worse fitting for the IR only; this is also consistent with added information provided by microwave channels in the AMSU instrument. Most of the globe is 0.1–0.2 higher for IR only, which is about 5% of the mean value of $\chi^2$=3-4 for retrievals restricted to QC=[0,1].

**Acknowledgments**

AMSR-E and AMSR-2 data products are produced by Remote Sensing Systems and were sponsored by the NASA AMSR-E Science Team, and all data are available through www.remss.com. The collocation methodology matching AIRS/AMSU with AMSR-E/AMSR-2 and DARDAR used index files generated through the NASA Earth Science Program NASA's Making Earth Science Data Records for Use in Research Environments (MEaSUREs) program. DARDAR-MASK and DARDAR-CLOUD are both available through the ICARE Thematic Center (http://www.icare. univ-lille1.fr/drupal/archive/). AIRS data were obtained through the Goddard Earth Services Data and Information Services Center (http://daac.gsfc.nasa.gov/). Brian H. Kahn was partially supported by the AIRS project at JPL and by the NASA Science of Terra and Aqua program under grant NNN13D455T. We thank two anonymous reviewers for very constructive comments on the manuscript, George Aumann, Steve Broberg, and Tom Pagano for detailed discussions about AIRS calibration, and Noel Cressie for guidance about testing for statistical significance. Part of this research was carried out at the Jet Propulsion Laboratory (JPL), California Institute of Technology, under a contract with the National Aeronautics and Space Administration. Copyright 2018. All rights reserved.

**Data citation**

Fetzer, E., Wilson, B., and Manipon, G.: AIRS-AMSU variables-CloudSat cloud mask, radar reflectivities, and cloud classification matchups V3.2, Greenbelt, MD, USA, Goddard Earth Sciences Data and Information Services Center (GES DISC), doi:10.5067/MEASURES/WVCC/DATA201, 2013.

Teixeira, J.: AIRS/Aqua L2 Standard Physical Retrieval (AIRS+AMSU) V006, Greenbelt, MD, USA, Goddard Earth
Sciences Data and Information Services Center (GES DISC), 10.5067/AQUA/AIRS/DATA201, 2013.

Wentz, F. J., T. Meissner, T., Gentemann, C., Brewer, M.: Remote Sensing Systems AQUA AMSR-E Daily Environmental Suite on 0.25 deg grid, Version V7. Remote Sensing Systems, Santa Rosa, CA. Available online at www.remss.com/missions/amsr, 2014. [Accessed 16 May 2017]

Wentz, F.J., Meissner, T., Gentemann, C., Hilburn, K. A., and Scott, J.:  Remote Sensing Systems GCOM-W1
AMSR2  Daily Environmental Suite on 0.25 deg grid, Version V7.2. Remote Sensing Systems, Santa Rosa, CA. Available online at www.remss.com/missions/amsr, 2014. [Accessed 17 Jul 2017]

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

|  | Mean Difference (Perturbed – Baseline IR/MW) | | Median Absolute Deviation (Perturbed – Baseline IR/MW) | |
|---|---|---|---|---|
|  | +50 mK | +50 mK before CC | +50 mK | +50 mK before CC |
| $r_{ei}$ (μm) | -0.105 | -0.131 | 0.786 | 1.168 |
| $r_{ei}$ AK | $-9.15 \times 10^{-5}$ | $+2.05 \times 10^{-6}$ | 0.00278 | 0.00247 |
| $\tau_i$ | +0.00534 | +0.0101 | 0.0812 | 0.149 |
| $\tau_i$ AK | -0.00117 | $-2.35 \times 10^{-4}$ | 0.0108 | 0.0053 |
| $T_{ice}$ (K) | +0.261 | +0.235 | 1.179 | 2.187 |
| $T_{ice}$ AK | $+4.73 \times 10^{-4}$ | +0.00113 | 0.00852 | 0.00999 |
| $\chi^2$ | -0.0176 | -0.0315 | 0.54 | 0.648 |

**Table 1. Shown are the two radiance perturbation tests after cloud clearing (+50 mK) and before cloud clearing (+50 mK before CC); see text for description), with reported mean difference and median absolute deviations of cloud properties, AKs, and $\chi^2$ for**

5    **54°S-54°N over ocean for 6 September 2002.**

|  | Mean Trend in Fig. 6 for IR only | | | Mean Difference (Perturbed – Standard IR/MW) | |
| --- | --- | --- | --- | --- | --- |
|  | SH | Tropics | NH | Steep | Shallow |
| $r_{ei}$ (μm) | +0.41 | +0.54 | +0.33 | -0.228 | +0.22 |
| $r_{ei}$ AK | -0.0012 | -0.0024 | -0.0009 | $+9.49 \times 10^{-4}$ | -0.00101 |
| $\tau_i$ | -0.066 | -0.019 | -0.079 | -0.0223 | +0.0233 |
| $\tau_i$ AK | +0.0031 | +0.0017 | +0.0023 | +0.00154 | -0.00154 |
| $T_{ice}$ (K) | -0.67 | -0.99 | -0.72 | -0.267 | +0.272 |
| $T_{ice}$ AK | -0.0018 | -0.0032 | -0.0011 | $-9.01 \times 10^{-4}$ | +0.00105 |
| $\chi^2$ | -0.009 | -0.019 | -0.0005 | -0.0161 | +0.0204 |

**Table 2. Shown are the trends depicted in Fig. 6 for the IR only retrieval, and the radiance perturbation sensitivity tests that adjust the slope in the retrieval channels. The 'steep' category indicates a steeper slope with retrieval channel 1 at 692.76 cm$^{-1}$ perturbed 50 mK cooler, retrieval channel 59 at 1133.91 cm$^{-1}$ perturbed 50 mK warmer, and a linear interpolation in between (see K14 for a list of the 59 retrieval channels). The 'shallow' category is reversed from the 'steep' category.**

|  | Upper Layer ECF | Lower Layer ECF | Total Counts | % of Total Counts |
|---|---|---|---|---|
| Opaque | ≥ 0.98 | Not specified | 890,747 | 1.90% |
| Non-Opaque | < 0.98 | < 0.1 | 34,310,209 | 73.42% |
| Multi-Layer | Not specified | ≥ 0.1 | 11,533,239 | 24.68% |

**Table 3. Definitions of the three ice cloud categories using the AIRS two-layer effective cloud fraction (ECF) product, the total counts and relative counts (%) of the three cloud categories. These results are obtained over the Tropical oceans (±18°) during July and August from 2003–2016.**

|             | LWP=0 RR=0 | LWP>0 RR=0 | LWP>0 RR>0 |
|-------------|------------|------------|------------|
| Opaque      | 12.9%      | 44.3%      | 42.8%      |
| Non-Opaque  | 23.0%      | 62.7%      | 14.3%      |
| Multi-Layer | 17.3%      | 53.3%      | 29.4%      |

**Table 4. The relative occurrence frequencies of non-zero liquid water path and rain rate according to coincident AMSR pixels for the three ice cloud categories defined in Table 3. These results are obtained over the Tropical oceans (±18°) during July and August from 2003–2016.**

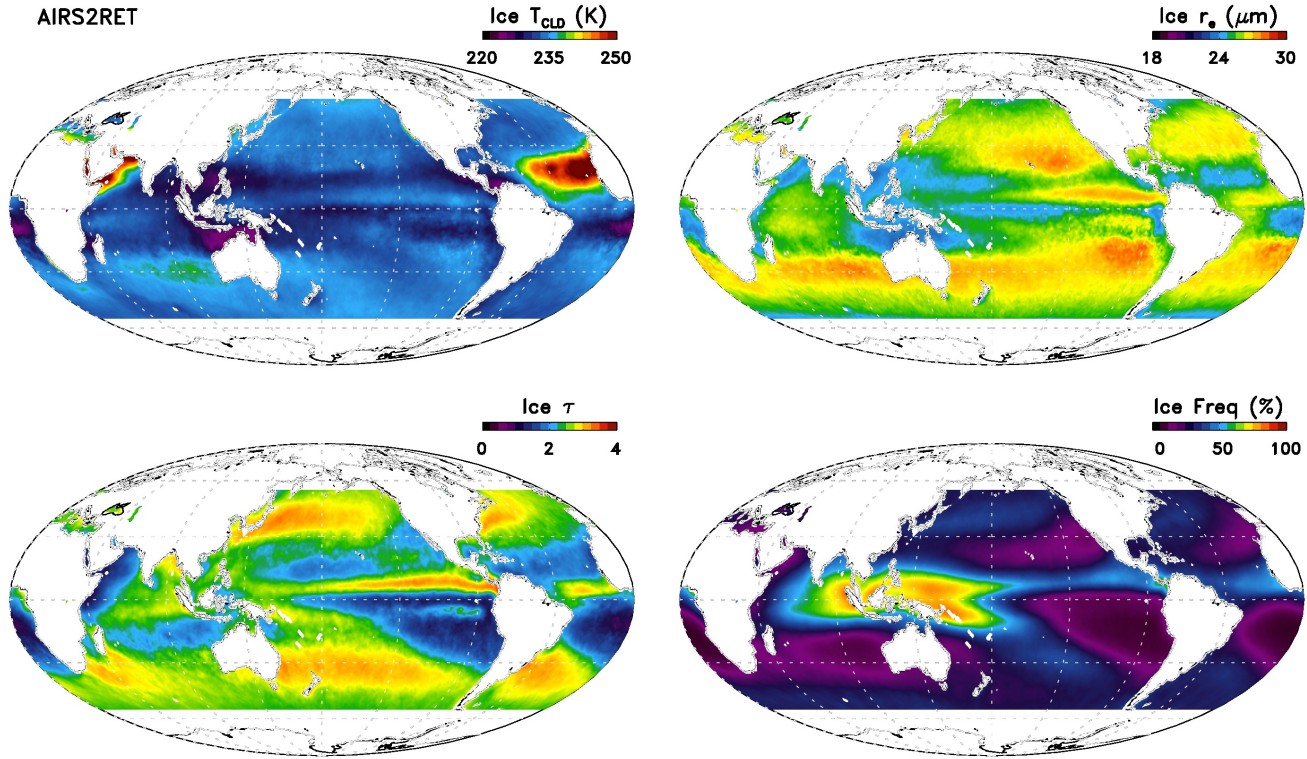

**Figure 1: Shown are the global 14-year averages (1 September 2002 –31 August 2016) of T_ci, r_ei, τ_i, and ice cloud frequency, for the AIRS IR only retrieval (AIRS2RET) between 54°S-54°N over the oceans. Note that the maximum T_ci value is higher than indicated in the color bar.**

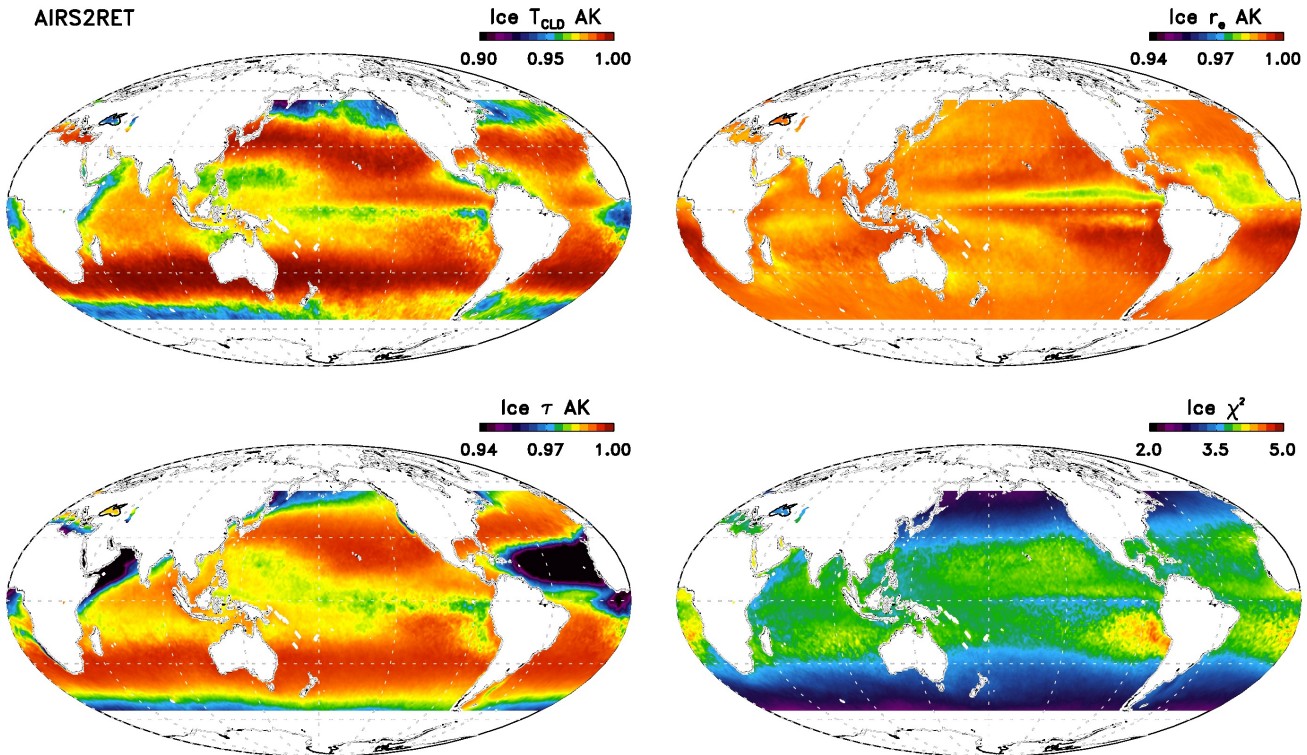

**Figure 2: Same as Fig. 1 except are the global 14-year averages of $T_{ci}$, $r_{ei}$, and $\tau_i$ averaging kernels (AKs) and $\chi^2$ fitting parameter. Note that the minimum AK values may be lower than indicated in the color bar.**

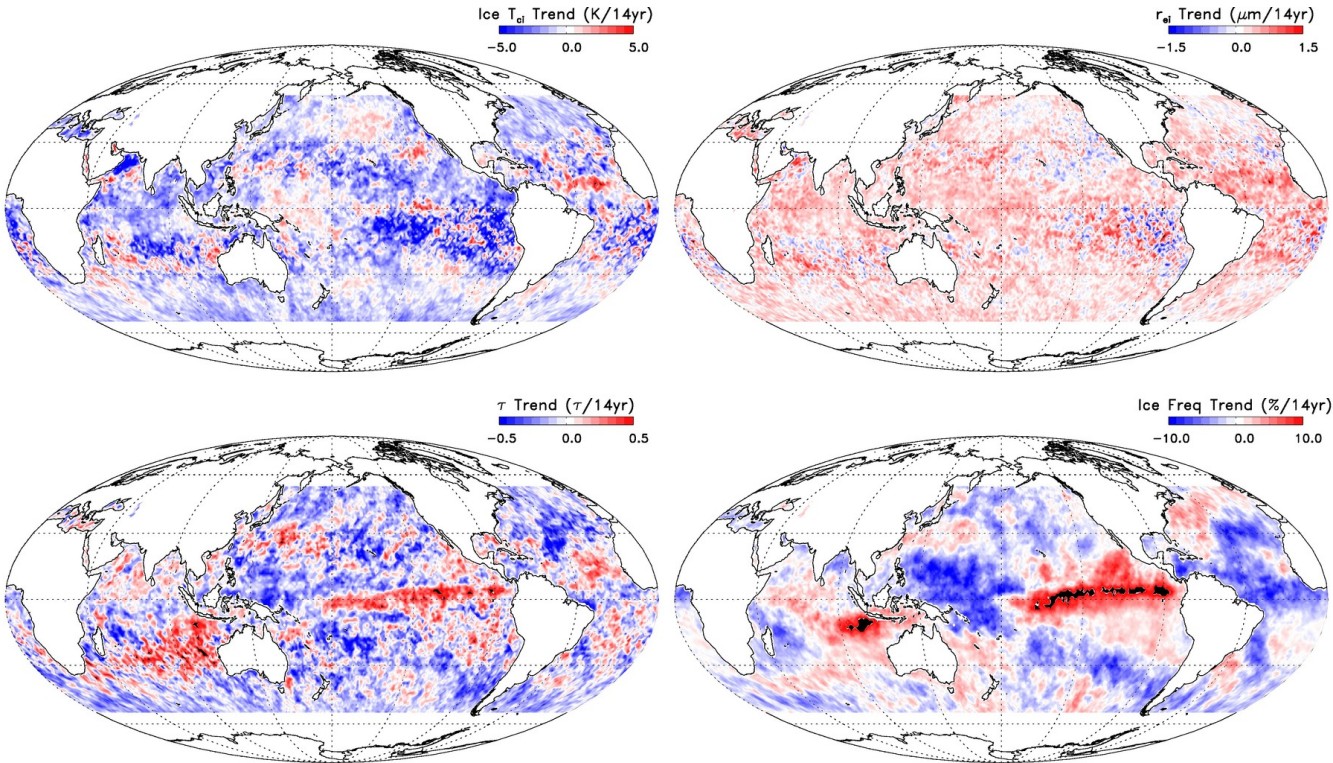

**Figure 3: Shown are the global 14-year trends for the fields in Fig. 1. Note that the minimum and maximum trends may exceed those indicated in the color bar.**

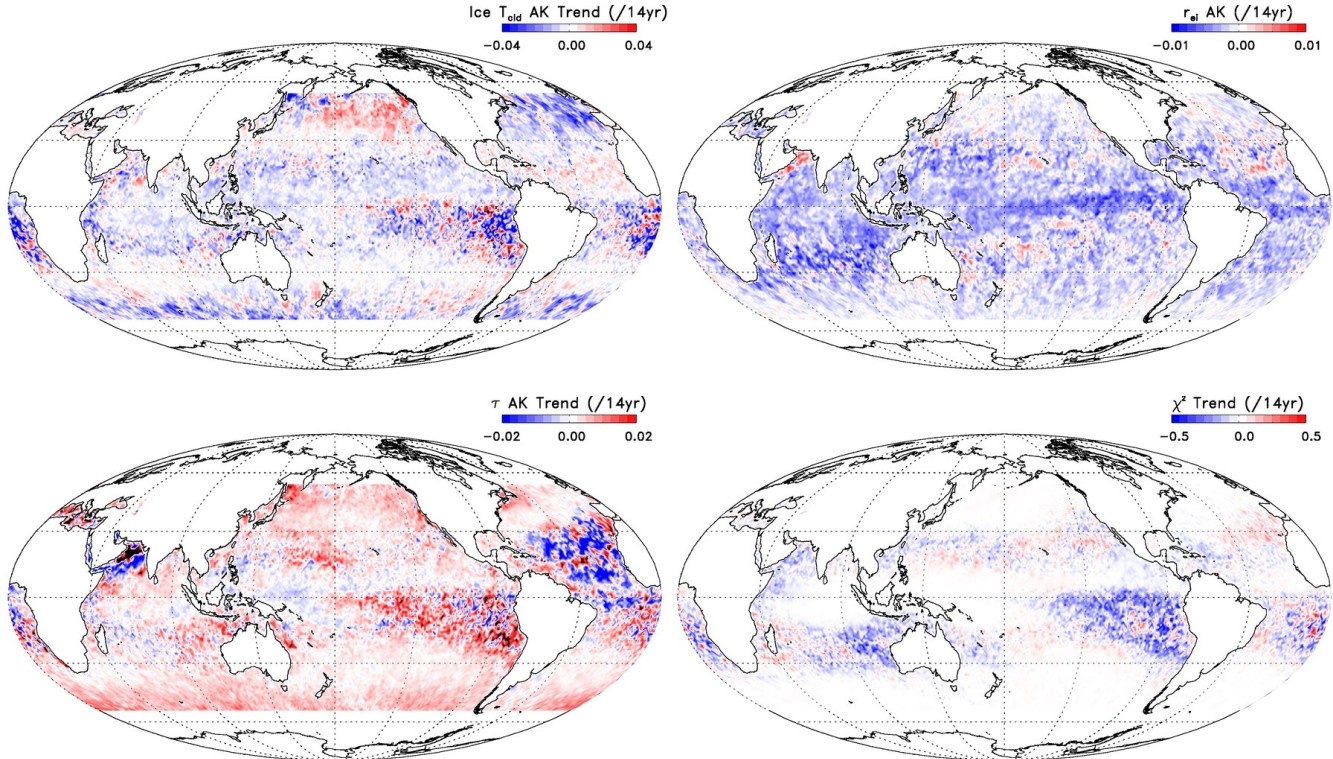

**Figure 4: Shown are the global 14-year trends for the fields in Fig. 2. Note that the minimum and maximum trends may exceed those indicated in the color bar.**

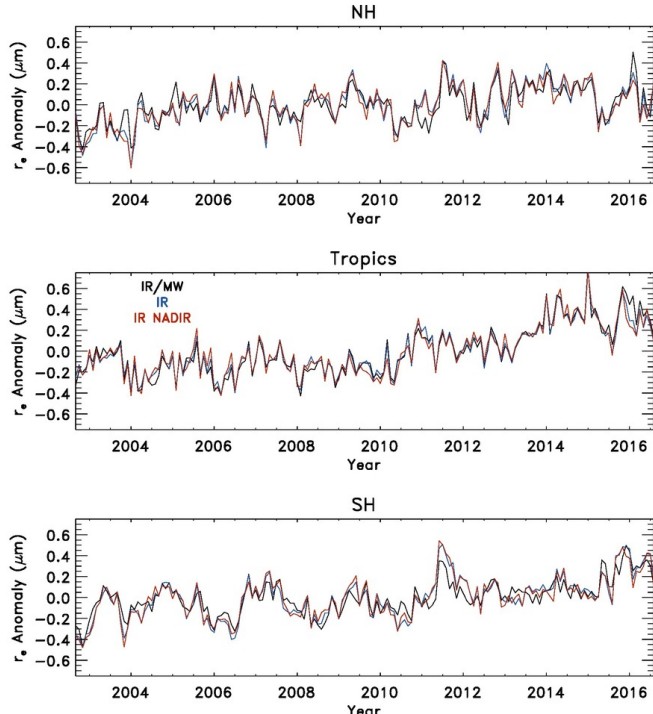

**Figure 5: Monthly anomaly time series for $r_{ei}$ for the IR/MW, IR only, and IR nadir retrievals organized into three latitude bands: SH extratropics (54°S-18°S), tropics 18°S-18°N, and NH extratropics (18°N-54°N).**

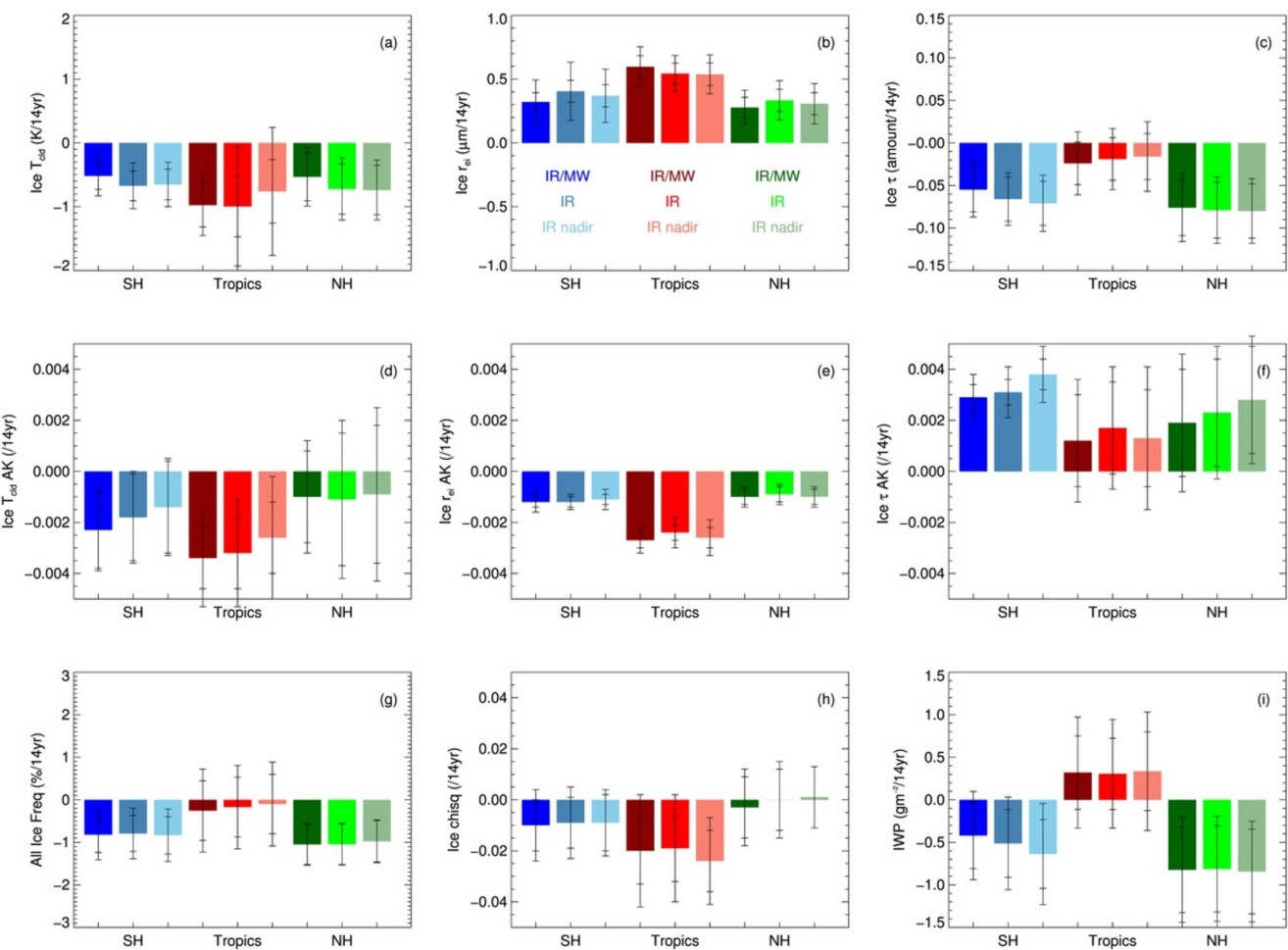

**Figure 6: Shown are the 14-year trends for the tropics, SH and NH extratropics, and IR/MW, IR only, and IR nadir retrievals. (a) $T_{ci}$, (b) $r_{ei}$, (c) $\tau_i$, (d) $T_{ci}$ AK, (e) $r_{ei}$ AK, (f) $\tau_i$ AK, (g) ice cloud frequency, (h) $\chi^2$, (i) IWP. The confidence for 95% statistical significance is shown as black lines with narrow tick marks and the confidence for 95% statistical significance with a lag-1 autocorrelation correction as wider tick marks (see Santer et al., 2000).**

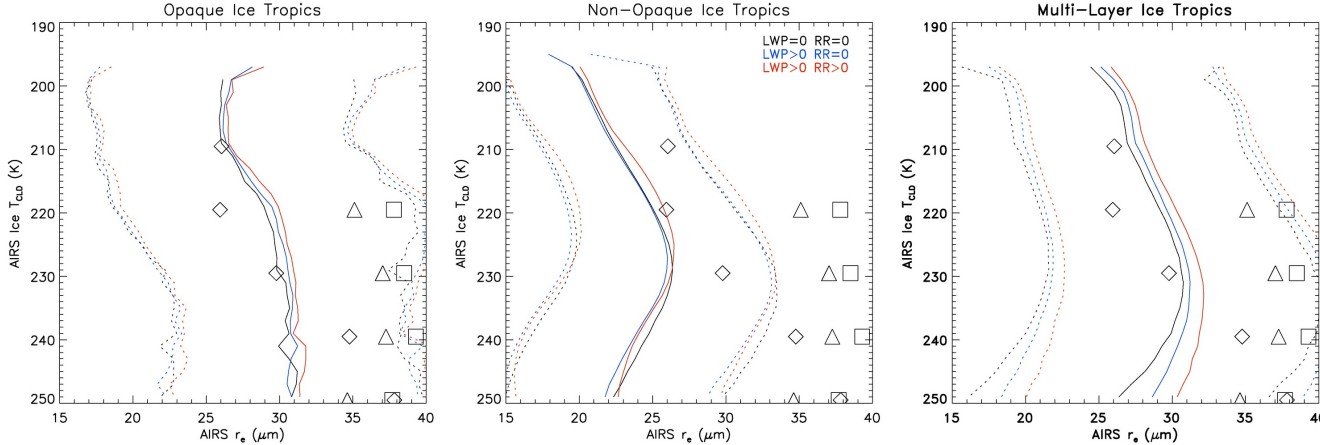

**Figure 7.** Shown are the mean (solid) and ±1σ values of AIRS IR only $r_{ei}$ versus $T_{ci}$ for opaque, non-opaque and multi-layer ice clouds over the tropical oceans during the months of July and August within the 14-year time period of investigation. Refer to Table 3 for definitions of the three categories. The AMSR-E/AMSR-2 estimates of LWP and RR are used to divide categories into clear (LWP=0) and no rain (RR=0) according to the passive microwave; cloudy (LWP>0) and no rain (RR=0); and cloudy (LWP>0) and rain (RR>0). The results in both panels have all AIRS ice clouds filtered within 6 K of the cold point tropopause; otherwise a much larger and potentially unphysical increase in $r_{ei}$ would be observed in the figures at the coldest temperatures. The symbols are average $r_{ei}$ from in situ field campaign data for the CRYSTAL-FACE (diamonds), TC4 (triangles), and NAMMA (squares) campaigns; please see Heymsfield et al. (2014) for more details.

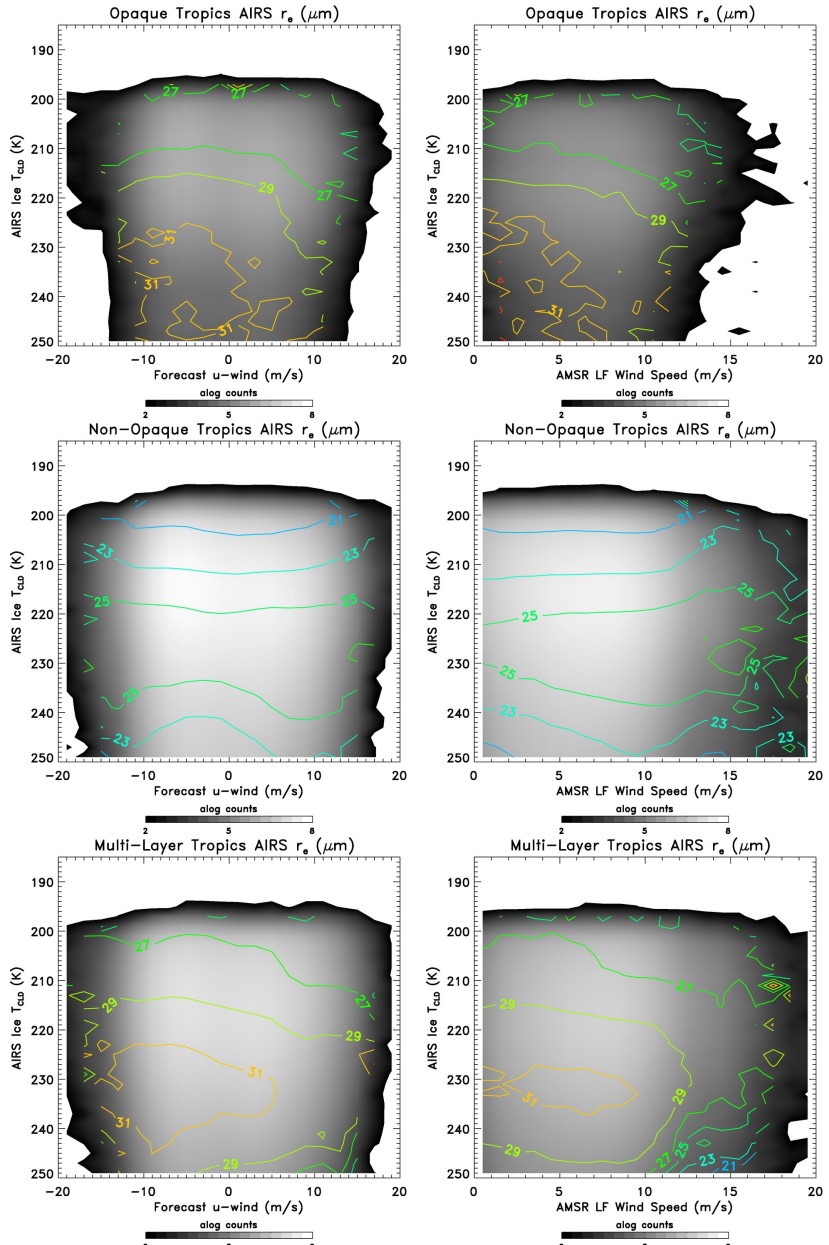

**Figure 8. Shown are AMSR-E/AMSR-2 low frequency (LF) wind speeds (m s$^{-1}$) (right column) or NWP model u-component wind speeds (left column) versus T$_{cld}$ histograms. The log counts shown are shown as gray scale, the AIRS IR only r$_{ei}$ (μm) overlaid as colored contours, with opaque (top row), non-opaque (middle row), and multi-layered (lower row) clouds shown separately.**

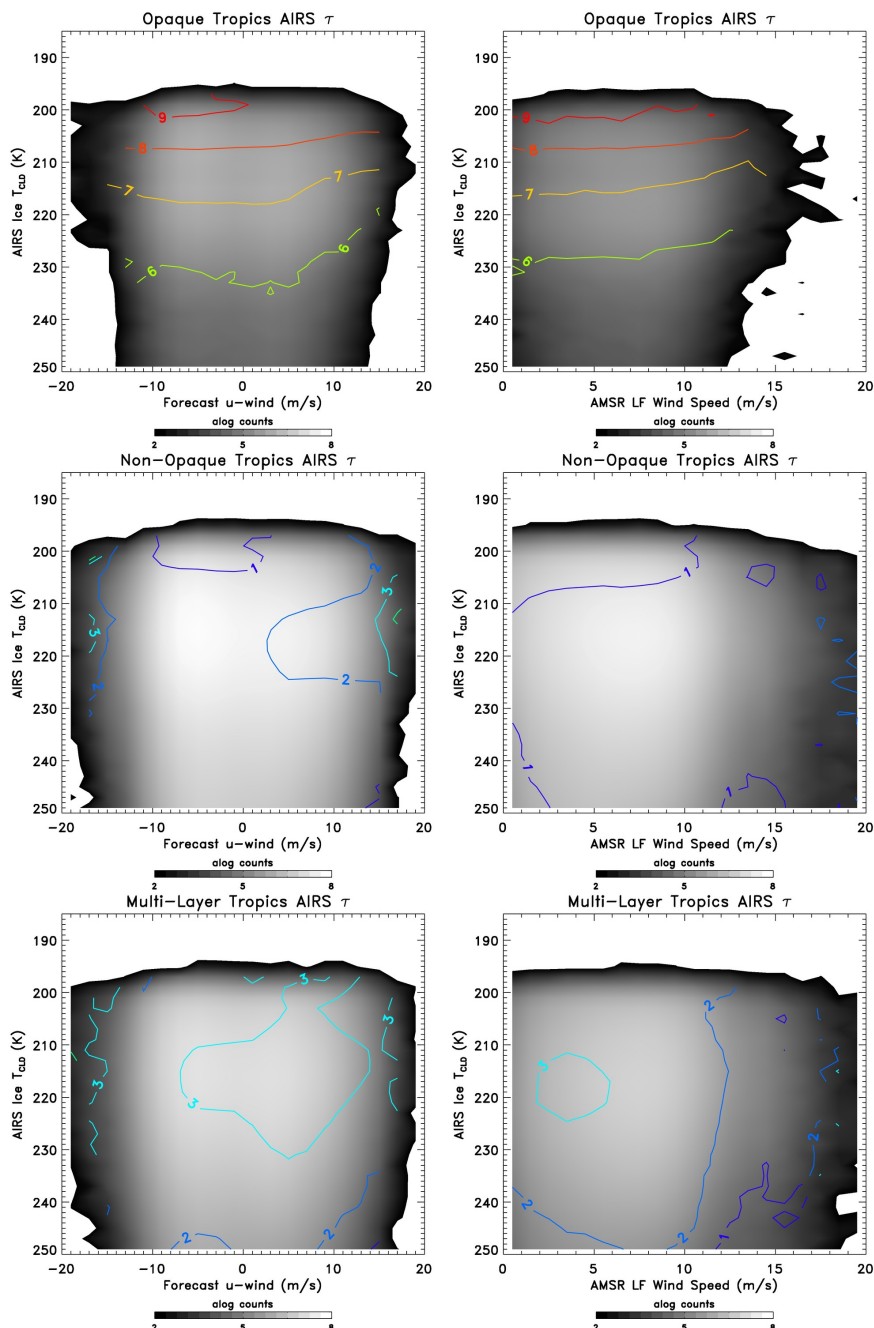

**Figure 9. Shown are AMSR-E/AMSR-2 low frequency (LF) wind speeds (m s$^{-1}$) (right column) or NWP model u-component wind speeds (left column) versus T$_{cld}$ histograms. The log counts shown are shown as gray scale, the AIRS IR only $\tau_i$ overlaid as colored contours, with opaque (top row), non-opaque (middle row), and multi-layered (lower row) clouds shown separately.**

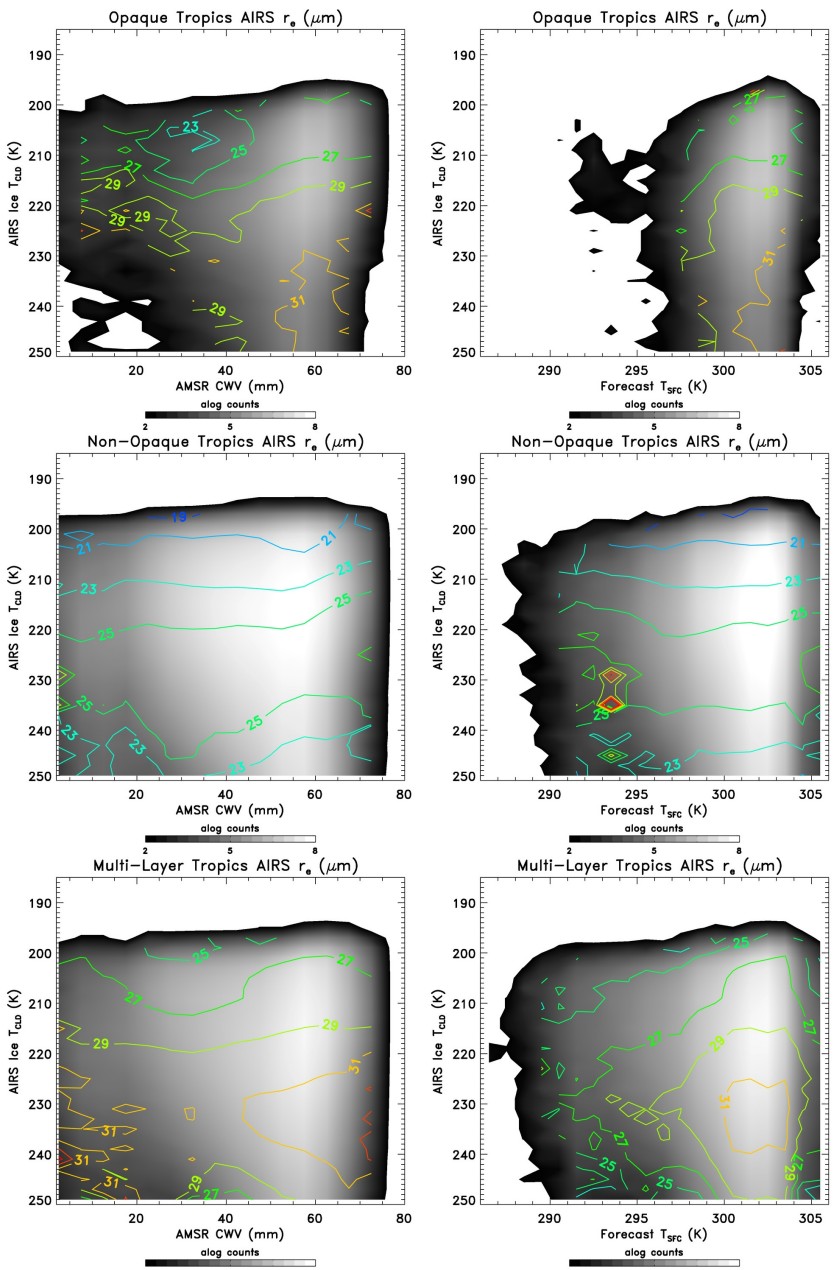

**Figure 10. Shown are AMSR-E/AMSR-2 total CWV (mm) (left column) or $T_{sfc}$ (K) (right column) versus $T_{cld}$ histograms. The log counts shown are shown as a gray scale, the AIRS IR only $r_{ei}$ (µm) overlaid as colored contours, with opaque (upper row), non-opaque (middle row), and multi-layered (lower row) clouds shown separately.**

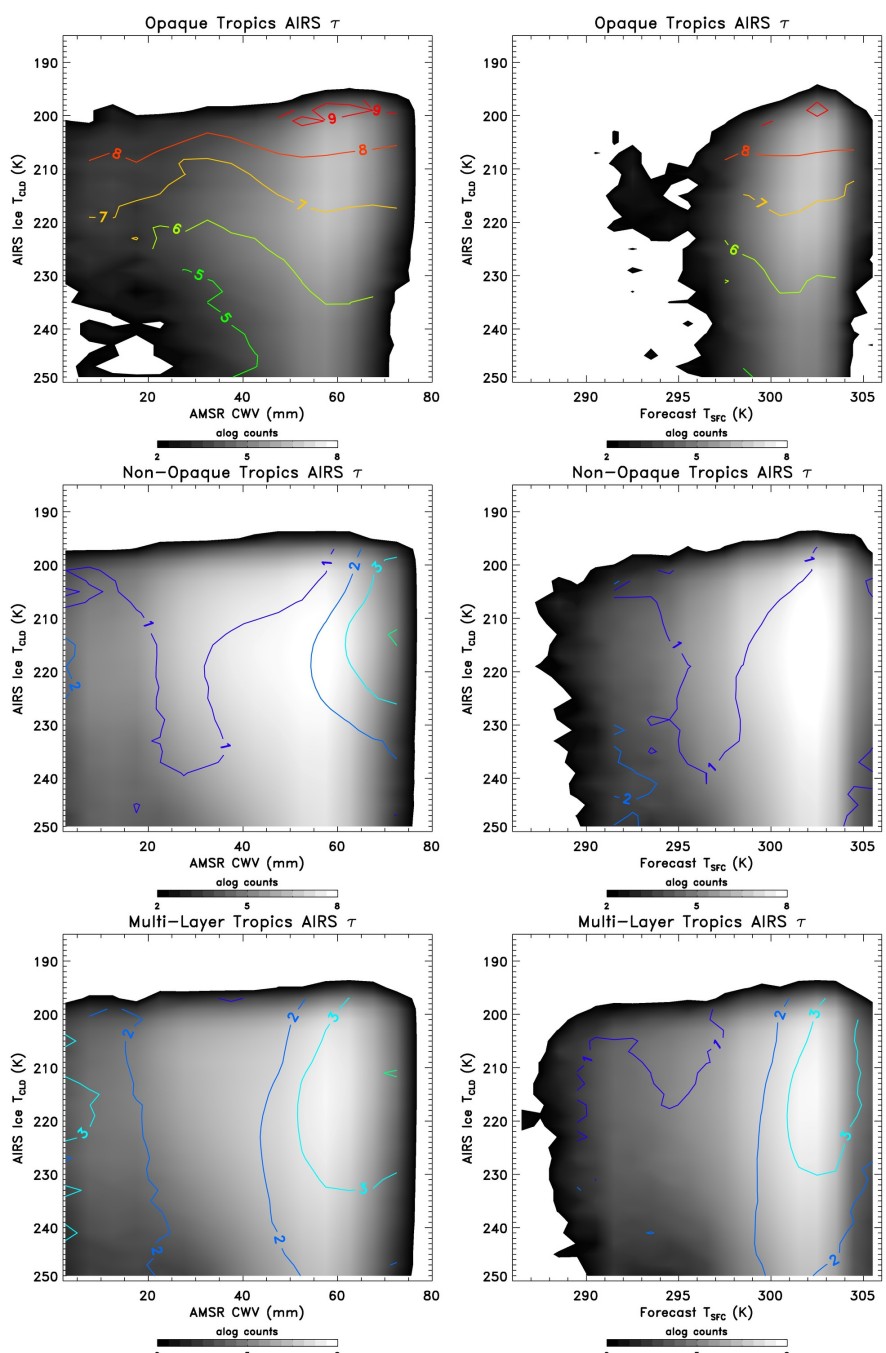

**Figure 11. Shown are AMSR-E/AMSR-2 total CWV (mm) (left column) or $T_{sfc}$ (K) (right column) versus $T_{cld}$ histograms. The log counts shown are shown as a gray scale, the AIRS IR only $\tau_i$ overlaid as colored contours, with opaque (upper row), non-opaque (middle row), and multi-layered (lower row) clouds shown separately.**

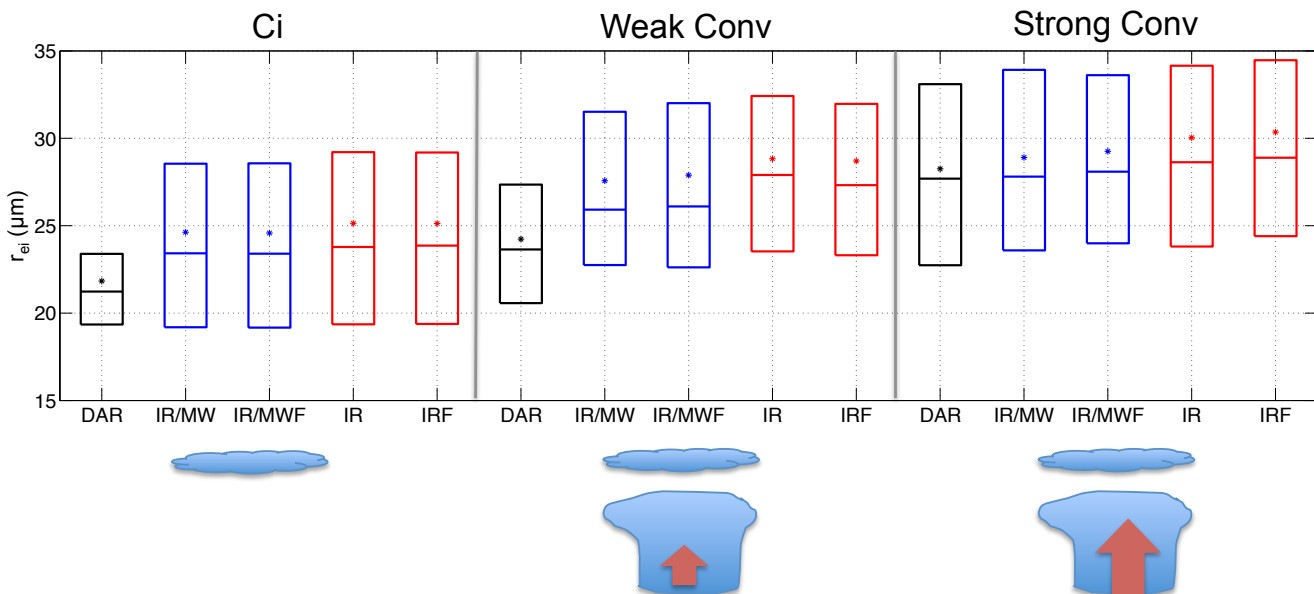

**Figure 12. DARDAR (black), IR/MW (blue) and IR only (red) retrievals of $r_{ei}$ in single-layered cirrus (left), cirrus on top of weak convection (center), and cirrus on top of strong convection (right); see Section 2.3 for a description of the definition of convective intensity. The AIRS retrievals are also shown for a filtered version that removes retrievals within 6 K of the tropopause. The central line is the median, the asterisk is the mean, and the bottom and top edges of the boxes are the 25th and 75th percentiles.**

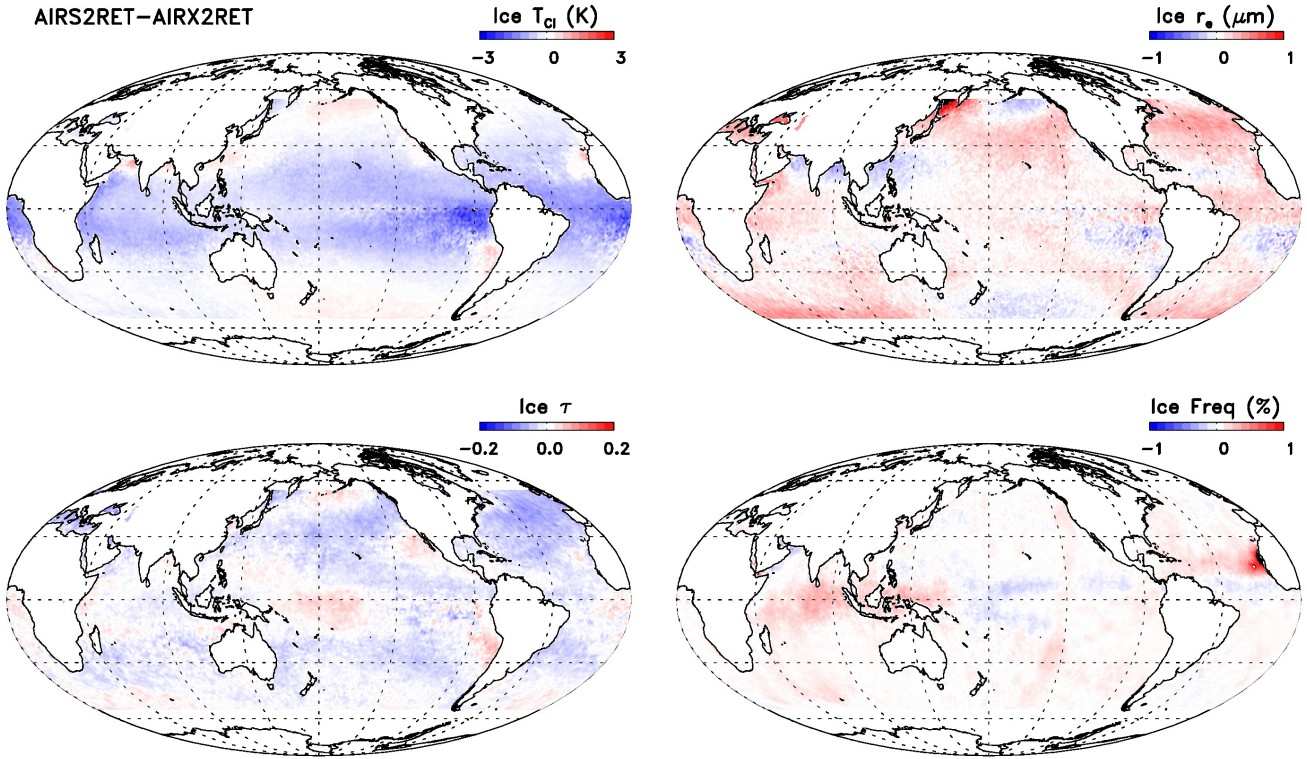

**Figure A1: Shown are the differences in IR only (AIRS2RET) minus IR/MW (AIRX2RET) for the same time period as described in Fig. 1 for $T_{ci}$, $r_{ei}$, $\tau_i$, and ice cloud frequency. Note that the minimum and maximum differences may exceed those indicated in the color bar.**

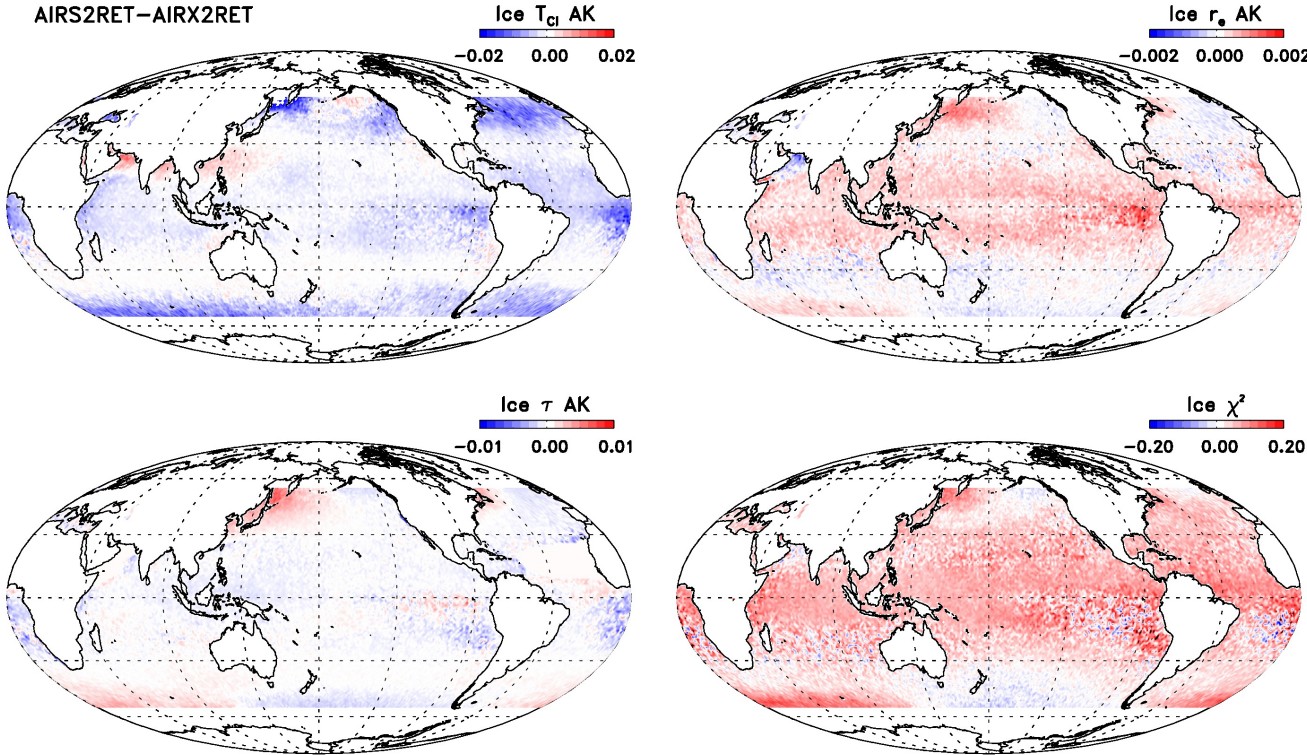

**Figure A2: Same as Fig. A1, except for the $T_{ci}$, $r_{ei}$, and $\tau_i$ AKs, and the $\chi^2$ fitting parameter. Note that the minimum and maximum differences may exceed those indicated in the color bar.**

