# Peer review of "Ice cloud microphysical trends observed by the Atmospheric Infrared Sounder"

_Atmospheric Chemistry and Physics, 2017_

## Referee Comment (RC1) · Anonymous Referee #2 · 22 Feb 2018

This is not a full review of this paper, but a request for clarification about the sampling of clouds by AIRS. I hope that the authors can reply to these questions promptly and at least before the end of the discussion period, so that I can adjust the full review accordingly. More information about the sampling is essential for interpreting the results and comparisons to earlier work.

It is not clear to me which clouds are included in the sample. The authors state that "AIRS sensitivity is maximized for optically thinner cirrus with $\tau \leq 5$". Given that the ice cloud properties are derived from infrared measurements, I interpret this as meaning that there is no information on optical depth or effective radius in these measurements for clouds with an optical thickness larger than 5. Yet, parts of the paper focus on convective clouds, of which larger parts would have optical thicknesses much larger

than 5. For example, figure 10 suggests that the properties of a cirrus above deep convection is retrieved. However, only the parts over thick outflow of that system would have column optical thickness lower than 5. If I am correct, AIRS would only sample the thick anvils of such clouds. Is this indeed the case?

I might be wrong though. Another interpretation is that there is no sensitivity to optical thickness for clouds with an optical thickness larger than 5, but there is still sensitivity to effective radius for these clouds in the AIRS wavelength range. In this case the sample would include essentially all clouds. The optical thickness would be 5 for any cloud thicker than that. Is this maybe the case?

Related to that, I am confused what sample of clouds are included in the 'opaque' cloud selection. Opaque is defined related to the effective cloud fraction, but it is unclear which optical thickness that would correspond to. If cloud with optical thicknesses larger than 5 are included in the sample, my guess is that these are opaque. If only clouds with optical thicknesses lower than 5 are included, what optical thickness range would correspond to 'opaque' clouds then?

---

## Author Comment (AC1) · 28 Feb 2018

Reviewer #2: This is not a full review of this paper, but a request for clarification about the sampling of clouds by AIRS. I hope that the authors can reply to these questions promptly and at least before the end of the discussion period, so that I can adjust the full review accordingly. More information about the sampling is essential for interpreting the results and comparisons to earlier work.

**Response: We thank the reviewer for prompting the authors for clarification. Please see below for our responses.**

Reviewer #2: It is not clear to me which clouds are included in the sample. The authors state that "AIRS sensitivity is maximized for optically thinner cirrus with tau < 5". Given that the ice cloud properties are derived from infrared measurements, I interpret this as meaning that there is no information on optical depth or effective radius in these measurements for clouds with an optical thickness larger than 5. Yet, parts of the paper focus on convective clouds, of which larger parts would have optical thicknesses much larger than 5. For example, figure 10 suggests that the properties of a cirrus above deep convection is retrieved. However, only the parts over thick outflow of that system would have column optical thickness lower than 5. If I am correct, AIRS would only sample the thick anvils of such clouds. Is this indeed the case?

**Response: The AIRS instrument is sensitive to tau <= 5 or so, but we want to clarify that this value is with respect to that defined in the infrared (about 11 microns). We report AIRS retrievals of tau with respect to 0.55 microns in the operational retrieval for easier comparison to the MODIS instrument. Please see Kahn et al., 2015, J. Geophys. Res. definitions and the comparisons to MODIS. We are able to extrapolate from 11 microns to 0.55 microns because we use Bryan Baum's bulk scattering models that are consistent across the wavelengths. In that case AIRS is sensitive to about tau <= 8 or so, with a few outliers that approach 10 (see Kahn et al., 2014, Atmos. Chem Phys., Figure 10 upper row).**

**As far as the sampling, we observe all clouds above some nominal tau > 0.1 or so, including convective clouds, but we are geometrically-speaking only sensing the upper tau <=5 (with respect to 11 microns). The same applies to the cloud effective radius (CER): it is retrieved for opaque/thick clouds only from the spectral signature in the upper 5 optical depths of the cloud. The rest of the cloud that is physically located below this upper layer is not retrieved since AIRS has no sensitivity to it.**

**So, bottom line, AIRS detects almost all ice clouds but the values of tau and CER are only obtained for the upper 5 optical depths.**

**In the revised manuscript, we will be clearer about the sensitivity of the AIRS instrument versus the sampling of the population of clouds.**

Reviewer #2: I might be wrong though. Another interpretation is that there is no sensitivity to optical thickness for clouds with an optical thickness larger than 5, but there

is still sensitivity to effective radius for these clouds in the AIRS wavelength range. In this case the sample would include essentially all clouds. The optical thickness would be 5 for any cloud thicker than that. Is this maybe the case?

**Response: The reviewer is correct that for very thick clouds the optical depth plateaus around 5-8. We include all cloud samples but the sensitivity limits us to the upper tau <=5 of the cloud. Furthermore, all CER retrievals are for the same portions of the cloud in which AIRS can sense tau.**

Reviewer #2: Related to that, I am confused what sample of clouds are included in the 'opaque' cloud selection. Opaque is defined related to the effective cloud fraction, but it is unclear which optical thickness that would correspond to. If cloud with optical thicknesses larger than 5 are included in the sample, my guess is that these are opaque. If only clouds with optical thicknesses lower than 5 are included, what optical thickness range would correspond to 'opaque' clouds then?

**Response: This is a great question. We did not report that in the submitted version of the manuscript, so we have included figures in this response for clarification. (We could add these figures, or similar ones, in the revision if the reviewer so chooses.) Below are figures 8 and 9 with ice cloud tau as a function of the AMSR variables and AIRS derived ice cloud top temperature, broken into opaque and transparent.**

**In general the opaque categories have several times larger values of tau than transparent. However, there is some structure in tau for some of the AMSR variables. Especially where the counts are lowest, tau can be below 5 for opaque clouds. This is not entirely surprising as lower layer clouds may exist and this drives the effective cloud fraction to near 1.0 even though some of the upper level ice cloud may in fact be transparent. The strong relationships of opaque tau with SST and column water vapor are very encouraging, and the dependence on ice cloud top temperature is also expected. The strong drop-off in tau with AMSR low frequency wind speed is quite interesting, but we note that the counts of those values are very low (the gray scale is on a log scale for counts). The asymmetry in tau with u-wind direction suggests larger tau for weak easterly winds. This is consistent with the arguments made, and cited literature to support the arguments made, in the manuscript about weak easterlies as more convectively active with somewhat larger cloud top CER values.**

---

## Referee Comment (RC2) · Anonymous Referee #2 · 6 Mar 2018

This is a review of the paper titled "Ice cloud microphysical trends observed by the Atmospheric Infrared Sounder" submitted to ACP by Kahn et al.

The paper presents retrievals of ice cloud properties using the AIRS instrument and discusses systematic variation of the ice cloud properties and trends seen in the record. The current paper mainly focuses on effective radius trends and variations. The paper is well written. The results are interesting and the techniques used are sound. In my opinion the results need to be related to past publications somewhat more. Also, as explained in my earlier online comment, the sampling of clouds is somewhat confusing, and makes interpretation difficult at times.

Firstly, I would like to thank the authors for replying to my first comments promptly. Those clarifications were helpful. I now understand that essentially all ice clouds

(tau>0.1) are included in the sample, the optical thickness values asymptotes at around 5, but for those thick clouds effective radius is still retrieved and included in the sample. Thank you in advance for making that clearer in the revised version of the paper.

I do have some more major comments or questions about the sampling of clouds in the paper. In additional, I have some minor comments. If the comments below are addressed in a revised version, I would recommend publication of this paper work in ACP.

1) In response to my question about what sample of clouds are included in the 'opaque' cloud selection, the authors included some more analysis in their reply. The included figures of the variation of optical thickness of opaque clouds appear to be interesting, although I'm still a bit confused about how to interpret them. The opaque clouds include clouds with optical thicknesses around 1 "as lower layer clouds may exist and this drives the effective cloud fraction to near 1.0 even though some of the upper level ice cloud may in fact be transparent." Thus, in such cases AIRS is able to retrieve the optical thickness of the upper ice cloud without interference of the lower clouds? If so, is the retrieved effective radius for these situations also not affected by the lower cloud?

Following this rationale, should I interpret retrievals of large optical thickness for opaque clouds as situations of thick ice clouds without any lower liquid clouds present to which AIRS is sensitive to. In turn, are retrievals of decreasing optical thickness for opaque clouds then associated with increasing interference of thick liquid clouds under the ice layers?

Please explain this further in the text. I suggest that the figures of the variation of optical thickness for opaque clouds are included in the paper and an interpretation of the systematic variation of ice cloud optical thickness with wind and surface temperature is provided.

2) The saturation of optical thickness at around 5 leads to a low bias of mean optical thickness for much of the globe, especially in the convectively active regions. Please

compare the global optical thickness distribution plots to those shown in King et al. (2013; reference below). King et al. (2013) report mean values generally exceeding 10, although the distribution is highly skewed towards thin clouds. Since the mean MODIS ice cloud optical thickness is dominated by the occurrence of thick clouds, the global distribution of MODIS optical thickness might be correlating better with the distribution of fraction of opaque clouds identified by AIRS. I suggest to include such a plot in the revised paper. Also, please note in section 4 of the paper that the saturation of optical thickness at around 5 also means that there is no sensitivity to any possible trends of ice cloud optical thickness of thicker clouds. The trend shown in Figures 5 and 6 are only reflecting trends in the optical thickness range for which AIRS is sensitive to. Finally, since trends in optical thickness may also lead to trends in the relative occurrence of opaque clouds, I suggest to also look at this and to describe the findings in the paper. Possibly a figure of the trends can be included in Figure 6.

3) Clouds closer than 6K to the cold point tropopause are removed from the sample for most part of the analysis. I wonder what potential influence this may have on the trends. In a warming world clouds may increase in height towards the tropopause over time. With this filtering in place, more cloudy pixels would be removed over time in that case. This may lead to an unrealistic positive bias in temperature trend and may also lead to biases on the mean effective radius and optical thickness trends. Since you are looking at rather small (but not unimportant!) trends, such sampling issues may impose relative large biases. Please investigate any possible trends in the filtered clouds and discuss it in the paper. Possibly a figure of the trends can be included in Figure 6.

Minor comments:

1) There seem to be issues with saturated colors in the global distribution plots. For example, in the Tcld plot in Figure 1, there is a white spot off the African coast that is surrounded by dark red colors. It seems that the white should be dark red. A similar thing happens in the trend plots, where regions that are off the scale on the low end are not dark blue, but white. Please inspect the plots for such occurrences and correct

the color scale.

2) In addition, please make the labels of the plots consistent with what is used in the rest of the papers (T_cld should be T_ci, etc.). Also, please check the text for consistency. At page 11, line 17, T_cld is used in the text, while otherwise T_ci is used, but there could be more of these inconsistent labelings.

3) At page 5, line 20, please provide reference(s) for the "well-documented spatial distributions of ice clouds". I suggest at least King et al. 2013. Also discuss the cloud fraction, height, optical thickness and effective radius distributions shown in Figure 1 in relation to those shown by King et al. 2013 and other relevant papers.

4) In section 4.2, trends on ice water path are introduced. Please write out "ice water path" before using the acronym. Also, please explain how IWP is determined. I suspect that is derived from the product of effective radius and optical thickness. That would mean that the absolute value is very much biased low for thick clouds because of the insensitivity to thick cloud optical thickness. Is the assumption here that the trends are not similarly affected?

5) Page 10, line 22: Van Diedenhoven et al. (2016) used airborne remote sensing data instead of in situ data.

6) Section 5.1: One of the three categories is where there is no cloud and no rain (CWP=0). This is confusing, since you are presenting cloud properties. Is "no cloud" really meaning no liquid part of the cloud? Please explain and change the nomenclature.

7) Figure 10: Please add "effective radius" to the y-axis label.

Reference: M. D. King, S. Platnick, W. P. Menzel, S. A. Ackerman and P. A. Hubanks, "Spatial and Temporal Distribution of Clouds Observed by MODIS Onboard the Terra and Aqua Satellites," in IEEE Transactions on Geoscience and Remote Sensing, vol. 51, no. 7, pp. 3826-3852, July 2013. doi: 10.1109/TGRS.2012.2227333

---

## Referee Comment (RC3) · Anonymous Referee #1 · 4 Apr 2018

This work investigates tendencies in the ice cloud frequency, effective radius (rei), optical depth (tau) and cloud top temperature in retrievals from the Atmospheric infrared sounder. The authors also analyze tendencies in the information content of the retrieval ruling out possible artifacts. Significant trends are found in the effective size, increasing over most of the globe, and also in other variables. The authors also show that rei is correlated with the column water vapor, and surface winds and temperature, particularly for opaque clouds suggesting an strong role of convection on the observed trends. Clouds are very sensitive to changes in the atmospheric state and trends in cloud properties may signal important systematic changes in Earth's climate. Hence this work is highly relevant to the atmospheric community and within the scope of ACP. The methods are sound and valid and the results interesting. I have some comments

on the organization of the work and in some places where more explanation and further analysis is required. After those are addressed this paper would be suitable for publication.

Comments: The paper reads a little bit like two separate works, with the second one starting in Section 5. Are the authors trying to use their analysis of convective processes to explain the observed tendencies? How do Figures 7-10 relate to the previous ones? I'd encourage the authors to work on making a consistent point throughout the paper.

How do the results compare to other instruments? They authors use MODIS and DARDAR in the analysis of the second part of the paper, but they seem to abstain of comparing the decadal trends from those products against their results. Are the tendencies from AIRS similar, at least qualitatively, to those of MODIS and DARDAR?

Page 11 Line 3. Please define the effective cloud fraction.

Section 5.1. The terms opaque and transparent seem ambiguous here. The authors draw a direct correlation between the covered area and the optical thickness of the clouds. But transparent cirrus (which most of the time refer to low optical depths) can be extended or simply cover a small fraction of the grid cell. Please clarify.

Section 5.2 and also in Section 5.3. Please add a paragraph explaining what you expect to see in the histograms (e.g., Figure 8) and how to interpret them. They are not obvious at all.

Page 11 Line 28. Here and in other places. It is not clear what the "reduction" is referring to. What is the control in this case? Please clarify.

---

## Author Comment (AC2) · 23 May 2018

This is a review of the paper titled "Ice cloud microphysical trends observed by the Atmospheric Infrared Sounder" submitted to ACP by Kahn et al. The paper presents retrievals of ice cloud properties using the AIRS instrument and discusses systematic variation of the ice cloud properties and trends seen in the record. The current paper mainly focuses on effective radius trends and variations. The paper is well written. The results are interesting and the techniques used are sound. In my opinion the results need to be related to past publications somewhat more. Also, as explained in my earlier online comment, the sampling of clouds is somewhat confusing, and makes interpretation difficult at times.

**We thank the reviewer for the very helpful and positive comments about the paper. Our responses to the reviewer comments are detailed below.**

Firstly, I would like to thank the authors for replying to my first comments promptly. Those clarifications were helpful. I now understand that essentially all ice clouds (tau>0.1) are included in the sample, the optical thickness values asymptotes at around 5, but for those thick clouds effective radius is still retrieved and included in the sample. Thank you in advance for making that clearer in the revised version of the paper.

**This is correct. We have essentially paraphrased a portion of your comment and inserted it into the revised manuscript in the middle of Section 2.1 (p.5, line 19 in the ACPD version): 'The AIRS sampling includes nearly all ice clouds with $\tau_i > 0.1$, while the maximum values of $\tau_i$ asymptote to values near 6-8 (e.g., Kahn et al., 2015). The $r_{ei}$ is retrieved for the same sample although retrievals with QC=2 are not included.'**

I do have some more major comments or questions about the sampling of clouds in the paper. In additional, I have some minor comments. If the comments below are addressed in a revised version, I would recommend publication of this paper work in ACP.

1) In response to my question about what sample of clouds are included in the 'opaque' cloud selection, the authors included some more analysis in their reply. The included figures of the variation of optical thickness of opaque clouds appear to be interesting, although I'm still a bit confused about how to interpret them. The opaque clouds include clouds with optical thicknesses around 1 "as lower layer clouds may exist and this drives the effective cloud fraction to near 1.0 even though some of the upper level ice cloud may in fact be transparent." Thus, in such cases AIRS is able to retrieve the optical thickness of the upper ice cloud without interference of the lower clouds? If so, is the retrieved effective radius for these situations also not affected by the lower cloud?

**We thank the reviewer for the insightful comments and careful reading of the manuscript. Given these comments and the ones that follow below, we have made some changes to the organization of the cloud categories. We now define three categories titled 'Opaque', 'Non-Opaque', and 'Multi-Layer' that are defined in a new Table 2. The Opaque category is restricted to an upper layer ECF >=0.98**

**(which implies that the lower layer ECF can be anywhere from 0.0 to 0.02). The Non-Opaque category is intended to capture approximately single layer clouds with an upper layer ECF < 0.98 and a lower layer ECF < 0.1.  (We decided to allow for some small amounts of lower layer cloud as this value has no material impact on the results except for moving around some of the pixels from one category to another.) The Multi-Layer category is intended to capture two-layer clouds with a lower layer ECF >= 0.1 with any value of upper ECF included in this category. This category shows the impacts of a lower layer of cloud on the retrievals or on the geophysical relationships shown in the joint histograms.**

**Thus, to answer the reviewer comments about a lower layer impacting the fidelity of the retrieval, that is entirely possible and those effects would in theory be exhibited more strongly in the Multi-Layer category.  Without going into a full sensitivity and retrieval impact study, however, we are unable to quantify the contributions of a second layer on CER and COT separately from the geophysical characteristics of this category, which may in fact be different than Opaque and Non-Opaque categories. However, with this additional category, we can to first order isolate the pixels that may be the most troublesome and are known to have somewhat lower values of information content in a multi-layer configuration (see Section 2). We have added the following text in Section 5.2 to clarify: '*Multi-layer clouds exhibit the largest changes with wind speed (Fig. 8). However, the reduced values of $r_{ei}$ at higher wind speeds have low frequencies of occurrence (i.e., noted by the gray scale shading). The contribution of retrieval biases that arise from an additional lower layer(s) not accounted for in the forward model (Kahn et al., 2014) has not been quantified. A firm conclusion on the realism of changes in multi-layer cloud top $r_{ei}$ to wind speed variability thus remains elusive and warrants further investigation.*'**

**Figure 7 now includes three panels, and Figures 8-11 now have six panels each, reflecting the Opaque, Non-Opaque, and Multi-Layer categories.  The original figures 8 and 9 for CER are now Figures 8 and 10.  The new Figures 9 and 11 are the new figures that contain the COT results. We point out that the Opaque variations seen in COT in the first response to reviewer #2 are in fact a result of clouds that included a significant magnitude of lower layer ECF.  In removing these and placing them into the Multi-layer category, the dependence of COT is much reduced, but is still non-zero, with respect to surface winds, CWV, and Tsfc.**

**We have added significant new discussion taking into context these three new categories and the new COT joint histograms and have substantially revised Sections 5.1 to 5.3. Please refer to track change version for specific changes as they are too numerous to list in the response.**

Following this rationale, should I interpret retrievals of large optical thickness for opaque clouds as situations of thick ice clouds without any lower liquid clouds present to which AIRS is sensitive to. In turn, are retrievals of decreasing optical thickness for opaque clouds then associated with increasing interference of thick liquid clouds under the ice layers?

**With the results emerging from the revised categories Opaque, Non-Opaque, and Multi-Layer, we believe that the reviewer interpretation is correct to first order. Now that the Opaque category does not contain significant lower layers of cloud that may contain liquid, we see that the COT has a stronger relationship with ice Tcld rather than with surface wind speed, CWV, Tsfc, etc. The COT histograms that were a part of the first reviewer response, which exhibit large gradients with wind speed, CWV, etc., have been reduced substantially or nearly eliminated in the new diagrams with three categories. Basically, the take home message is that cloud vertical structure and cloud regime exhibit different signals, and if mixed together in the same diagram, can lead to the interpretation of gradients that do not show up in individual cloud regimes.**

Please explain this further in the text. I suggest that the figures of the variation of optical thickness for opaque clouds are included in the paper and an interpretation of the systematic variation of ice cloud optical thickness with wind and surface temperature is provided.

**We have included the optical thickness histograms for all of the panels depicted in Figures 8 and 9, which are now Figures 8 and 10. The COT versions of the histograms are now Figures 9 and 11. All edits are included in the track change version and are found in Sections 5.1 to 5.3.**

2) The saturation of optical thickness at around 5 leads to a low bias of mean optical thickness for much of the globe, especially in the convectively active regions. Please compare the global optical thickness distribution plots to those shown in King et al. (2013; reference below). King et al. (2013) report mean values generally exceeding 10, although the distribution is highly skewed towards thin clouds. Since the mean MODIS ice cloud optical thickness is dominated by the occurrence of thick clouds, the global distribution of MODIS optical thickness might be correlating better with the distribution of fraction of opaque clouds identified by AIRS. I suggest to include such a plot in the revised paper.

**The suggestion of an additional plot showing correlations between optical thickness among AIRS and MODIS for optically thicker clouds is well taken. In Kahn et al., (2015), J. Geophys. Res., MODIS and AIRS are compared for one month of data using pixel-level comparisons sorted by scene complexity. We quote from Kahn et al (2015): '*For four positive (ice phase) tests, an approximately zonal symmetric pattern emerges and is similar to MODIS ice cloud $\tau$ distributions described in King et al. [2013]. This result is encouraging as it demonstrates the overlapping sensitivity of subsets of AIRS and MODIS ice cloud properties for thicker ice clouds.*' Instead of reproducing material from a previous paper, we decided to point the reader to Kahn et al. (2015) paper where a very detailed comparison between MODIS and AIRS is described. We have added text following that added above describing AIRS sampling: '*Kahn et al. (2015) describe pixel-level comparisons between AIRS and MODIS ice cloud properties and show that overlapping sensitivity for both $\tau_i$ and $r_{ei}$***

*are maximized for optically thicker pixels containing four positive ice phase tests with spatial maps resembling those described in King et al. (2013).'*

Also, please note in section 4 of the paper that the saturation of optical thickness at around 5 also means that there is no sensitivity to any possible trends of ice cloud optical thickness of thicker clouds. The trend shown in Figures 5 and 6 are only reflecting trends in the optical thickness range for which AIRS is sensitive to.

**We have added the following text in the beginning of Section 4.1 to clarify: '*We reiterate that the AIRS sensitivity to ice clouds is limited between 0.1 < $\tau_i$ < ~6-8, thus the $\tau_i$ trends do not include potential trends outside of this sensitivity range.*'**

Finally, since trends in optical thickness may also lead to trends in the relative occurrence of opaque clouds, I suggest to also look at this and to describe the findings in the paper. Possibly a figure of the trends can be included in Figure 6.

**This is a good question and is something we are currently working towards answering. We intend on submitting a separate manuscript that addresses trends in ECF, Tcld, thermodynamic phase partitioning in the extratropics, and tie these results together to the present work, which is focused on ice microphysics. The trends in opaque clouds are somewhat ambiguous. We are still trying to determine the best approach to calculate these trends, which variables to use, and how to filter and/or classify the data. We have calculated the trends for ECF with regard to all ice clouds and are included as Fig. 3 of this response (Figs. 1 and 2 are in the response to reviewer #1). The trends are not statistically significant at the 95% level.**

3) Clouds closer than 6K to the cold point tropopause are removed from the sample for most part of the analysis. I wonder what potential influence this may have on the trends. In a warming world clouds may increase in height towards the tropopause over time. With this filtering in place, more cloudy pixels would be removed over time in that case. This may lead to an unrealistic positive bias in temperature trend and may also lead to biases on the mean effective radius and optical thickness trends. Since you are looking at rather small (but not unimportant!) trends, such sampling issues may impose relative large biases. Please investigate any possible trends in the filtered clouds and discuss it in the paper. Possibly a figure of the trends can be included in Figure 6.

**The reviewer's comments above are well taken and we did not discuss this in the manuscript. We have repeated the trend analysis for no 6 K filtering within the tropopause and the results are included as Figs. 4 and 5 in this response for $\tau_i$ and $r_{ei}$, respectively. The results are virtually identical to those shown in Fig. 6 in the submitted manuscript. The tropopause filtering has no material impact on the sign, magnitude, and statistical significance of the zonal band trends. All other variables shown in Fig. 6 in the submitted manuscript yield very similar results upon manual inspection but we did not include them here because of prioritization of the reviewer response subject matter and author time constraints. To address this point in the revised manuscript, we have included the following statement in Section 4.2: '*The**

*results in Fig. 6 have clouds within 6 K of the tropopause filtered out; very similar results are obtained with no filtering, and no material changes in sign, magnitude, and statistical significance are found.'*

Minor comments:
1) There seem to be issues with saturated colors in the global distribution plots. For example, in the Tcld plot in Figure 1, there is a white spot off the African coast that is surrounded by dark red colors. It seems that the white should be dark red. A similar thing happens in the trend plots, where regions that are off the scale on the low end are not dark blue, but white. Please inspect the plots for such occurrences and correct the color scale.

**In these cases the values went beyond the color scale. These have been fixed. We point out in the figure caption that areas at either end of the color scale may contain values lower or greater than the color scale indicates. Instead of stretching the color scale to minimum and maximum values, we prefer not focusing on the outliers and instead show the more interesting structure in a more narrow range.**

2) In addition, please make the labels of the plots consistent with what is used in the rest of the papers (T_cld should be T_ci, etc.). Also, please check the text for consistency. At page 11, line 17, T_cld is used in the text, while otherwise T_ci is used, but there could be more of these inconsistent labelings.

**This is a good catch that requires some additional explanation. Figures 7-11 use the upper level cloud top temperature Tcld from the standard cloud clearing retrieval on the y-axis. Tcld is used as the prior guess to Tci (additional text to clarify on p. 4). We have made identical plots with Tci and they are generally very similar but there can be a few changes especially in the 230-250 K range. Recall that Tci is derived assuming a single layer ice cloud while Tcld is derived for up to two layers; thus the upper layer Tcld is judged to be somewhat more precise in multi-layer clouds (see Yue et al., 2017b, AIRS Version 6 Test Report). We are currently working on a follow-on manuscript that addresses these and other cloud variables other than ice microphysics. We will discuss and reconcile the different cloud top temperature estimates and their trends, as this topic deserves a separate venue for discussion.**

**We have added the following text to Section 3.3 to explain the choice of the y-dimension: '*As discussed in Kahn et al. (2014), the $T_{ci}$ variable is included in the retrieval state vector to improve the chi-square radiance fits and the success rate of retrieval convergence. While there are strong similarities between $T_{ci}$ and the upper level $T_{cld}$, some differences arise within multi-layer clouds as expected since $T_{ci}$ is based on the assumption of a single-layer cloud (Kahn et al., 2014). Further discussion on the reconciliation of the two cloud top temperatures is in progress and will be presented in a separate manuscript.'***

3) At page 5, line 20, please provide reference(s) for the "well-documented spatial distributions of ice clouds". I suggest at least King et al. 2013. Also discuss the cloud fraction, height, optical thickness and effective radius distributions shown in Figure 1 in

relation to those shown by King et al. 2013 and other relevant papers.

**We added King et al. (2013), Wylie and Menzel (2005), and Stubenrauch et al. (2013) to point the reader to standard references for ice cloud property distributions from a variety of satellite remote sensing data sets.**

4) In section 4.2, trends on ice water path are introduced. Please write out "ice water path" before using the acronym. Also, please explain how IWP is determined. I suspect that is derived from the product of effective radius and optical thickness. That would mean that the absolute value is very much biased low for thick clouds because of the insensitivity to thick cloud optical thickness. Is the assumption here that the trends are not similarly affected?

**We now spell out ice water path (IWP) at first usage in section 4.2. The IWP is calculated using a standard relationship between $\tau_i$ and $r_{ei}$ and have included a brief description of the calculation: '*The IWP is calculated using the relation IWP = (2/3) $\rho_i$ $\tau_i$ $r_{ei}$, where $\rho_I$ is the density of ice (0.92 g cm$^{-3}$).*' Indeed, the trends for the optically thick clouds are likely biased low. We refer reviewer #2 to the response to reviewer #1 regarding using other data sets. We showed trends for MODIS in the response, and in fact, the COT trends are larger in the convective areas compared to AIRS, but the patterns are very similar. This topic deserves further examination but is well beyond the scope of the present work as we discussed in the reply to reviewer #1.**

5) Page 10, line 22: Van Diedenhoven et al. (2016) used airborne remote sensing data instead of in situ data.

**Fixed.**

6) Section 5.1: One of the three categories is where there is no cloud and no rain (CWP=0). This is confusing, since you are presenting cloud properties. Is "no cloud" really meaning no liquid part of the cloud? Please explain and change the nomenclature.

**Since passive microwave radiometry is sensitive to LWP and not IWP, we describe in the text that this is due to liquid condensate and not ice. We have changed all uses of CWP to LWP in the text, figures, and tables.**

7) Figure 10: Please add "effective radius" to the y-axis label.

**Fixed.**

Reference: M. D. King, S. Platnick, W. P. Menzel, S. A. Ackerman and P. A. Hubanks, "Spatial and Temporal Distribution of Clouds Observed by MODIS Onboard the Terra and Aqua Satellites," in IEEE Transactions on Geoscience and Remote Sensing, vol. 51, no. 7, pp. 3826-3852, July 2013. doi: 10.1109/TGRS.2012.2227333

[Figure]

**Figure 3. AIRS ECF trends for all ice clouds for the three latitude bands and three sampling/algorithm categorizations shown in Fig. 6 of the manuscript.**

[Figure]

**Figure 4. AIRS ice cloud COT trends for the three latitude bands and three sampling/algorithm categorizations shown in Fig. 6 of the manuscript. No 6 K filtering is applied. The results are nearly identical to the 6 K filtering version.**

[Figure]

**Figure 5. AIRS ice cloud CER trends for the three latitude bands and three sampling/algorithm categorizations shown in Fig. 6 of the manuscript. No 6 K filtering is applied. The results are nearly identical to the 6 K filtering version.**

---

## Author Comment (AC3) · 23 May 2018

This work investigates tendencies in the ice cloud frequency, effective radius (rei), optical depth (tau) and cloud top temperature in retrievals from the Atmospheric infrared sounder. The authors also analyze tendencies in the information content of the retrieval ruling out possible artifacts. Significant trends are found in the effective size, increasing over most of the globe, and also in other variables. The authors also show that rei is correlated with the column water vapor, and surface winds and temperature, particularly for opaque clouds suggesting an strong role of convection on the observed trends. Clouds are very sensitive to changes in the atmospheric state and trends in cloud properties may signal important systematic changes in Earth's climate. Hence this work is highly relevant to the atmospheric community and within the scope of ACP. The methods are sound and valid and the results interesting. I have some comments on the organization of the work and in some places where more explanation and further analysis is required. After those are addressed this paper would be suitable for publication.

**We thank the reviewer for the positive comments about the paper. Our responses to the reviewer comments are included below.**

Comments: The paper reads a little bit like two separate works, with the second one starting in Section 5. Are the authors trying to use their analysis of convective processes to explain the observed tendencies? How do Figures 7-10 relate to the previous ones? I'd encourage the authors to work on making a consistent point throughout the paper.

We agree that the paper would benefit from additional clarification regarding how the different sections are connected to each other. To address this concern, we have made the following edits to the manuscript:

We have added the following statement as the last line of the last paragraph in the Introduction: '*The collocation of pixel-scale data among AIRS, AMSR, and DARDAR is a first step towards illuminating the potential processes that may be responsible for secular changes in ice cloud properties.*'

To link earlier sections with AMSR-AIRS joint histograms, we have added the following sentence to the end of the paragraph in Section 3.3: '*Convective and non-convective cloud types are shown separately in order to highlight the much larger responses of rei to thermodynamic and dynamical variability in tropical convection.*'

To link earlier sections with AIRS-DARDAR results, we have added the following sentence to the end of the paragraph in Section 3.4: 'As mentioned in Section 3.3, the coarse classification of convective intensity helps highlight differences in cloud top rei for non-convective, weakly, and strongly convective scenes.'

In the first paragraph of Section 5, we have replaced the last line with the following text to emphasize that dynamic and thermodynamic variability lead to changes in rei and hence offer possible reasons for trends in rei: *Collectively, the*

aforementioned modeling and observational investigations suggest that rei varies significantly at the tops of convective ice clouds and motivates the synergistic use of AMSR and AIRS at the pixel scale to capture convective-scale processes.'

We have modified the second to last line in the first paragraph of Section 6 to tie together AIRS-DARDAR to convective processes and observations of trends: 'Systematic changes in the global circulation and changes in convective clustering and cloud overlap may lead to a higher frequency of overlapping cirrus on top of convection, and a reduced frequency of thin cirrus with climate change evidenced by trends in the Multi-angle Imaging SpectroRadiometer (MISR) derived cloud texture (Zhao et al., 2016).'

We have added the following statement as the second to last line in the second to last paragraph of Summary and Conclusions: '*The pixel-level collocations of AIRS, AMSR, and DARDAR are a first attempt at identifying atmospheric processes that could be responsible for secular trends in ice cloud properties.*'

How do the results compare to other instruments? They authors use MODIS and DARDAR in the analysis of the second part of the paper, but they seem to abstain of comparing the decadal trends from those products against their results. Are the tendencies from AIRS similar, at least qualitatively, to those of MODIS and DARDAR?

We appreciate the reviewer suggestion to include other data sets to assess trends. We ultimately decided not to focus on data sets from other instruments for multiple reasons. First, an assessment of the retrieval algorithms and stability of e.g. the MODIS radiances/reflectances, CloudSat reflectivity, and CALIOP/IIR backscatter/radiances would require significant research to assess suitability for trend analysis, and the expertise of the lead author is limited to AIRS instrumentation. This is well beyond the scope of the current investigation, but it is reasonable to expect that in the future this may be possible in collaboration with multiple instrument/algorithm teams. Second, with regard to MODIS cloud retrieval data, the lead author has been in contact with the MODIS team (personal communication) and they are currently performing a similar analysis for MODIS Aqua to infer cloud property trends, including ice clouds. With recent advancements in MODIS calibration, they are now pushing harder on quantifying trends. We defer to the expertise on the MODIS team to determine suitability for trend analysis. Third, the CloudSat temporal record begins in July 2006, the spatial sampling is restricted to near nadir view in the AIRS swath, and the sensitivity to thin cirrus is limited. (Similar limitations are noted for CALIPSO, but may be more relevant to AIRS comparisons because of excellent sensitivity to thin cirrus.) Thus, the DARDAR product (which combines CloudSat, CALIPSO, and MODIS) ultimately contains these spatial and temporal sampling limitations, and possible radiometric drift uncertainties.

Despite the paragraph above, the lead author has calculated trends in MODIS CER and COT using the Collection 6, Level 3, monthly gridded data. The anomaly time

series is calculated using the same approach and software routine for AIRS data. The results for the same 14-year period are shown below in two panels. The level of agreement in the spatial patterns of COT, even for fairly small-scale features that could be interpreted as noise, is quite remarkable between AIRS and MODIS. Note that the trend magnitude in the color scale for MODIS is an order of magnitude larger than AIRS, however. With regard to CER, MODIS shows more expansive negative regions than AIRS and the patterns are rather dissimilar in various regions. There may be geophysical, algorithmic, cloud type sensitivity, sampling, and additional reasons for this but a robust explanation is well beyond the scope of this article. The lead author (Kahn) has agreed to work with the MODIS team in the near-term to reconcile these differences once the MODIS team has published the results of their trend study.

Page 11 Line 3. Please define the effective cloud fraction.

We have added the following text to the next line in the same paragraph: '*The ECF* is a cloud product that represents the convolution of cloud fraction and cloud emissivity. Nasiri et al. (2011) showed that ECF from AIRS and effective emissivity from MODIS is in excellent agreement for both single and multi-layered cloud configurations.'

Section 5.1. The terms opaque and transparent seem ambiguous here. The authors draw a direct correlation between the covered area and the optical thickness of the clouds. But transparent cirrus (which most of the time refer to low optical depths) can be extended or simply cover a small fraction of the grid cell. Please clarify.

We agree with the reviewer. After further thought about the naming convention, we have decided to change 'Transparent' to 'Non-Opaque', which implies either full coverage of a pixel by a transparent cloud, or partial coverage of a pixel by optically thick clouds, or somewhere in between. We have made the text changes in the manuscript, figure captions, and the figures themselves.

We also note that suggestions by reviewer #2 led us to define a third category of cloud configuration called 'Multi-Layer'. We have added a new Table 3 that defines these categories and have edited the first paragraph of Section 5.1 to clarify these definitions. Please refer to the response to reviewer #2 for a detailed explanation of our related changes to the revised manuscript.

Section 5.2 and also in Section 5.3. Please add a paragraph explaining what you expect to see in the histograms (e.g., Figure 8) and how to interpret them. They are not obvious at all.

We have added the following text to the first paragraph in Section 3.3: '*The* histograms each contain one of several AMSR variables on the x-axis and the AIRS upper layer  $T_{cld}$  on the y-axis.'

We have added substantial revisions to Section 5.1 to 5.3 that address both reviewer concerns about interpretation and reviewer #2's concerns about lower-layer clouds. Please refer to the response to reviewer #2 for additional detail.

Page 11 Line 28. Here and in other places. It is not clear what the "reduction" is referring to. What is the control in this case? Please clarify.

In this instance, we have changed 'reduction' to 'weaker vertical dependence'. In Section 2.1, we have changed a few instances from 'reduction' to 'lower magnitude' or 'minima'. In Section 4.2, we have changed 'reduction' to 'decrease'. In Section 5.2, we have changed 'reduction' to 'change'. In Section 7, we have changed 'reduction' to 'weaker dependence'. All of these changes are included in the track change version.

Figure 1. MODIS ice COT trends for the same 14-year period shown in the manuscript Fig. 3 for AIRS.

Figure 2. MODIS ice CER trends for the same 14-year period shown in the manuscript Fig. 3 for AIRS.